# Heat Stress and Plant–Biotic Interactions: Advances and Perspectives

**DOI:** 10.3390/plants13152022

**Published:** 2024-07-23

**Authors:** Rahul Mahadev Shelake, Sopan Ganpatrao Wagh, Akshay Milind Patil, Jan Červený, Rajesh Ramdas Waghunde, Jae-Yean Kim

**Affiliations:** 1Division of Applied Life Science (BK21 Four Program), Plant Molecular Biology and Biotechnology Research Center, Gyeongsang National University, Jinju 52828, Republic of Korea; 2Global Change Research Institute, Czech Academy of Sciences, Brno 60300, Czech Republic; cerveny.j@czechglobe.cz; 3Cotton Improvement Project, Mahatma Phule Krishi Vidyapeeth (MPKV), Rahuri 413722, India; akshaypatilbiotech@gmail.com; 4Department of Plant Pathology, College of Agriculture, Navsari Agricultural University, Bharuch 392012, India; rajeshpatho191@nau.in; 5Division of Life Science, Gyeongsang National University, Jinju 52828, Republic of Korea; 6Nulla Bio Inc., Jinju 52828, Republic of Korea

**Keywords:** biotic stress, climate change, climate-resilient crops, heat stress, microbiome, plant immunity, plant stress, plant–biotic interactions

## Abstract

Climate change presents numerous challenges for agriculture, including frequent events of plant abiotic stresses such as elevated temperatures that lead to heat stress (HS). As the primary driving factor of climate change, HS threatens global food security and biodiversity. In recent years, HS events have negatively impacted plant physiology, reducing plant’s ability to maintain disease resistance and resulting in lower crop yields. Plants must adapt their priorities toward defense mechanisms to tolerate stress in challenging environments. Furthermore, selective breeding and long-term domestication for higher yields have made crop varieties vulnerable to multiple stressors, making them more susceptible to frequent HS events. Studies on climate change predict that concurrent HS and biotic stresses will become more frequent and severe in the future, potentially occurring simultaneously or sequentially. While most studies have focused on singular stress effects on plant systems to examine how plants respond to specific stresses, the simultaneous occurrence of HS and biotic stresses pose a growing threat to agricultural productivity. Few studies have explored the interactions between HS and plant–biotic interactions. Here, we aim to shed light on the physiological and molecular effects of HS and biotic factor interactions (bacteria, fungi, oomycetes, nematodes, insect pests, pollinators, weedy species, and parasitic plants), as well as their combined impact on crop growth and yields. We also examine recent advances in designing and developing various strategies to address multi-stress scenarios related to HS and biotic factors.

## 1. Introduction

Climate change, a substantial risk to crop productivity globally, undermines food security. Rising global temperatures often coincide with droughts, leading to significant declines in crop yields [1,2,3]. In addition to agricultural sustainability, biodiversity is at risk due to global climate change, leading to extreme weather events, which disturb ecosystems and ultimately give rise to multiple social issues [1,4,5]. On a worldwide scale, the predicted effects of the combined stressors (biotic and abiotic factors) will result in an annual yield loss of more than 50% for major agricultural commodities (summarized in Appendix A) [6]. The frequent occurrence of high temperature-mediated heat stress (HS) challenges plant survival by reducing water availability. Also, rising temperatures are predicted to accelerate the incidence and severity of plant diseases and epidemics [7,8]. Particularly, HS exacerbates plant pathogens, amplifying infections across diverse plant species [9]. Host biodiversity, spatial organization, and abiotic factors significantly impact biotic diseases and are rapidly changing due to climate change, habitat loss, and changes in nitrogen deposition [10]. This effect is most visible in certain parts of the world, predominantly in Asian and African nations, where various climate-related extremes, such as droughts, heat waves, unpredictable rainfall, storms, floods, and the emergence of pests, have negatively impacted the livelihoods of farmers [2,11].

The optimal temperature plays a critical role in plant growth and development, impacting crop productivity and the timing of cropping seasons. Temperatures exceeding the optimal range are recognized as HS in all living organisms. Climate models predict further temperature rises and unpredictable rainfall patterns with increased intensity in the coming days. While current models offer potential weather predictions of the near future, frequent temperature fluctuations resulting from climate change pose a significant challenge to their accuracy [12]. Variability in predicting climate extremes and knowledge gaps in our understanding of how plants defend themselves against combinatorial pressures require systematic studies to gain molecular and physiological insights, ultimately seeking sustainable solutions for such new-age agricultural issues [13,14]. Novel approaches towards climate-ready crops are needed to address this climate change scenario.

Meanwhile, crop susceptibility to multiple environmental stresses may result from extensive domestication and selective breeding, resulting in the loss of alleles associated with tolerance to abiotic and biotic stresses [15]. Wild cultivars survive under multi-stress conditions because of their diverse microbiome. Most studies focus on investigating plant–microbe interactions in controlled environmental conditions like greenhouses or growth chambers, typically aiming at only one aspect of the study. Such experimental data provide a limited understanding of the dynamics of plant–pathogen–environment interactions [16]. Environmental factors are known to drive plant resistance pathways and influence pathogen virulence. Understanding the HS effect on plant physiology and associated microbiome and their responses to plant pathogens can be the basis of developing such climate-resilient crops for a better future.

This review explores the influences of plant HS on interactions with major biotic factors, including fungi, bacteria, viruses, phytoplasma, nematodes, parasitic weeds, insect pests, and pollinators. Understanding these interactions is crucial for developing agricultural strategies to mitigate the impacts of climate change on crop yields, enhance plant resilience, and ensure global food security. Moreover, there is a pressing need to comprehend the plant microbiome’s role under concurrent stress conditions. Here, we underscore the imperative of exploring the interplay between heat and biotic factors, elucidating their effects on plant molecular and physiological responses. Emphasizing the microbiome’s significance in coping with combined stressors, we advocate for strategies to develop multi-stress-resilient crop varieties and sustainable agricultural practices, leveraging beneficial microbes and modern tools.

## 2. Heat Stress and Plant–Biotic Interactions

The plant kingdom encompasses species adapted to nearly every ecosystem on Earth, enduring temperature variations ranging from about −80 °C to 70 °C [17]. Every plant species, including cultivated crops, exhibits an optimal temperature range or threshold that directly influences its growth, development, and yields (Figure 1A). The optimal temperature range varies depending on several factors, including the plant species, specific developmental stage, and environmental conditions such as light, humidity, and soil type. At each growth stage, from germination to flowering and beyond, there exists an ideal temperature range within which the plant thrives and performs effectively to the best of its genetic makeup [18]. Deviations from this range can result in suboptimal growth, reduced yield, or adverse effects on the plant’s health and productivity.

Current agriculture faces a threat of combinational stress scenarios exacerbated by climate change issues. Also, plant breeding efforts have enabled the development of crop varieties that are tolerant to individual stress factors and have achieved high yield and vigor capabilities through crop modification via breeding and domestication. However, changes in recurring patterns of abiotic factors like heat, moisture, drought, and light can affect plants and pests, altering plant susceptibility to pathogens and the extent of damage caused by pests [19,20,21]. HS and biotic stress are particularly interlinked and involve several unexplored facets that occur simultaneously or sequentially, reducing crop yields (Figure 1B). Recent studies have highlighted the dual challenges plants face with simultaneous biotic stresses, such as pathogens and pests, alongside the HS [19,22]. 

Systematic studies are necessary to understand the impact of concurrent HS and biotic stressors on crop productivity [23,24]. Increased temperatures leading to HS directly correlate with varying (positive or negative) effects on plant defenses, resulting in higher susceptibility or tolerance to biotic stressors (Figure 1B). These effects depend on the order of stress occurrence, plant stage, genetic background, and environmental factors [24,25,26]. Long-term exposure to HS is generally expected to affect plant health negatively. Thus, a thorough knowledge of how HS influences the physiological and molecular mechanisms involved in plant immune responses is essential. This knowledge can enhance crop productivity and aid in developing climate-resilient crop varieties. However, most research in this domain has focused on individual stresses, with relatively little exploration of combined stressors [7,27]. Therefore, the effects of HS and biotic stress on plants need to be studied under individual stress conditions and combined stresses, as elaborated in the subsequent sections. First, we provide an overview of the individual responses to HS and biotic factors. Subsequently, we elaborate on plant responses to HS–biotic factor interactions under multi-stress scenarios.

## 3. Plant Responses to Heat Stress

When subjected to HS, plants face numerous challenges and respond with various adaptive strategies to minimize damage and ensure survival, as illustrated in Figure 2. These responses involve adaptations in growth, development, and physiology, coordinated through chemical and hormonal signaling synchronized with the differential expression of stress-responsive genes at the molecular level. HS notably affects critical plant processes such as transpiration, photosynthesis, membrane thermostability, respiration, and osmoregulation at the subcellular, cellular, and whole-plant levels [28]. As summarized in Figure 2, the challenges faced by plants ultimately impact growth and significantly reduce plant productivity [29,30]. Plant morphological changes resulting from temperature deviations below the HS range and outside the optimal range are collectively termed thermomorphogenesis [31,32]. Thermomorphogenesis leads to elongated hypocotyls, elongated petioles, reduced stomata, and smaller and thinner leaves, facilitating surface cooling [32]. 

Plants have evolved multiple ways to escape, avoid, or tolerate HS. These mechanisms involve maintaining water balance, enhancing antioxidant defenses, altering hormonal signaling, repairing membranes, and balancing ion levels. Additionally, plants make metabolomic and genomic adjustments to adapt to HS. For example, chloroplasts play a crucial role in photosynthesis and temperature sensing, thereby initiating retrograde signaling-mediated suitable physiological responses [33]. HS often reduces photosynthetic efficiency, primarily due to the impairment of photochemical reactions in the thylakoid lamellae and carbon metabolism in the chloroplast stroma. Also, the disruption of cell membranes increases permeability and compromises normal cellular functions [34]. 

Reactive oxygen species (ROS) production and accumulation increases under HS in various cell organelles, including chloroplasts, mitochondria, cell walls, peroxisomes, the plasma membrane, the apoplast, and the endoplasmic reticulum (ER) [30]. Major ROS include hydroxyl radicals, superoxide, singlet oxygen, and hydrogen peroxide [3]. Excessive levels of ROS induce oxidative stress, leading to membrane damage, protein degradation, and enzyme inactivation, thereby reducing plant viability [35,36]. Furthermore, HS alters plant water status, causing dehydration and impacting growth and development [28,37]. Stomatal regulation is one of the primary HS responses adopted by plants to conserve water and cool leaves to prevent impairment of photosynthesis processes. Plants close their stomata to reduce water loss through transpiration while activating antioxidant defense systems to scavenge harmful ROS produced under HS. However, prolonged exposure to high temperatures necessitates stomatal opening to cool plant leaves through transpiration [38]. 

To protect plant cells from oxidative damage caused by ROS, plants synthesize various molecules for ROS detoxification and scavenging. Along with the stimulated production of secondary messengers or signaling molecules [e.g., Calcium (Ca^2+^), ROS, nitric oxide, trace elements, polyamines, lipids, and hydrogen sulfide], phytohormones also emerged as major contributing factors that activate various metabolic pathways and genetic machinery governing HS responses (summarized in Figure 2). Phytohormones implicated in plant HS include gibberellic acid (GA), abscisic acid (ABA), salicylic acid (SA), jasmonic acid (JA), ethylene (ET), brassinosteroids (BR), and cytokinins (CK) [37,39,40,41]. For instance, ABA controls stomatal closure and modulates the expression of several stress-responsive genes to facilitate adaptation to HS [39]. 

Molecular responses to HS tolerance in plants involve the induction of two major groups of proteins: heat shock transcription factors (HSFs) and heat shock proteins (HSPs) [18]. HSPs act as cellular chaperones, helping to stabilize proteins under stress. Among the HSFs, HSFA1 plays a central role in regulating the expression of other HSFs, thereby providing HS tolerance to plants [42]. HSFA2 is also a key HS regulator targeted by HSFA1 and governs the expression of HSP-mediated HS responses [5]. Recently, Kan and colleagues [5] systematically summarized the HS-responsive genetic networks in crops and the model plant *Arabidopsis*. Their summary highlights the crucial role of HSF and HSP genes through transcriptional regulation, signaling molecules, non-coding RNAs (ncRNAs), epigenetic modifications (including chromatin remodeling, histone modification, and DNA methylation), unfolded protein responses (UPRs) in the ER, and other genetic changes in the chloroplast and mitochondrial genomes. These diverse responses underscore the complex mechanisms plants use to cope with HS, emphasizing the importance of understanding physiological and molecular aspects for crop improvement.

## 4. Plant Responses to Biotic Stressors

Plants face constant threats from various biotic agents (summarized in Figure 1B). To counter these threats, plants have developed a complex immune system consisting of two interconnected layers: pattern-triggered immunity (PTI) and effector-triggered immunity (ETI) (Figure 3). Microbe-, pathogen-, nematode-, herbivore-, and parasite-associated molecular patterns (MAMPs, PAMPs, NAMPs, HAMPs, and ParAMPs) trigger PTI. Pathogen- and parasite-derived effectors (e.g., bacterial flagellin, lipopolysaccharide, peptidoglycan, fungal chitin, viral components, and nematode pheromones) are perceived by host-derived immunogenic molecules in ETI and PTI [43,44,45,46,47].

The host plant-derived molecules include pathogenesis-related resistance (PR) proteins, pattern-recognition receptors (PRRs), and their interacting co-receptors (Co-PRRs), nucleotide-binding domain leucine-rich repeat receptors (NLRs), danger- or damage recognition molecules (damage-associated molecular patterns, DAMPs), R proteins encoded by *R* genes, and immunomodulatory phytocytokines [48,49,50,51]. The PR protein group includes plant proteins induced by molecules from defense signaling pathways activated following a phytopathogenic attack [49]. The accumulation of PR proteins is triggered by SA, JA, and other defense-related pathways, which helps the plant reduce the pathogenic load and prevent the spread to non-infected plant parts. There are 19 families of PR proteins classified as PR-1 to PR-19 based on their biochemical nature [52]. Some of these PR proteins have enzymatic activity, such as β-glucanases (PR-2), chitinases (PR-3, PR-8, and PR-11), peroxidases (PR-9), and ribonucleases (PR-10). PR proteins promote innate resistance in plants by breaking down fungal cell walls, making membranes permeable, suppressing transcription, and deactivating ribosomes [53]. Recent literature summarizes previous research highlighting the crucial role of PR proteins in plant resistance to phytopathogens, making them promising candidates for developing disease-resistant crop varieties [52,53]. Although PR proteins are also implicated in abiotic stress responses, their biochemical basis is not entirely known and needs further studies.

Phytocytokines are peptides secreted by plants in response to infections, functioning as secondary signals during both biotic and abiotic stresses and playing roles in plant growth and development [54]. Accumulating evidence suggests that PTI and ETI trigger a cascade of pathways involving overlapping signaling modes and immune responses mediated by phytohormones and secondary messengers. Generally, PRRs, DAMPs, and phytocytokines activate PTI, while NLRs recognize effectors to activate ETI. However, recent literature highlights that recognizing extracellular effectors by PRRs blurs the exact distinction between PTI and ETI. NLR receptors can detect intracellular effector molecules to active ETI responses in three ways: either through direct interaction, by recognizing changes in a decoy protein that mimics the structure of an effector, or by perceiving alterations in the cytosolic domains of PRRs caused by extracellular effectors [55,56] (Figure 3). Receptor-like kinases (RLKs) and their associated cytoplasmic members, receptor-like cytoplasmic kinases (RLCKs), and RLK–RLCK modules are the primary PRR components. Together, they regulate various processes in response to plant growth, development, and biotic and abiotic stimuli [57]. The recognition of pathogenic signals (elicitors) or effectors by the plant immune system activates multiple phosphorylation-mediated signaling pathways, including ROS signaling and Ca^2+^ channels. Activation of PTI and ETI triggers various immune responses, such as transcriptional reprogramming, activation of Ca^2+^-dependent protein kinases (CDPKs) and mitogen-activated protein kinase (MAPK) cascades, ROS-mediated oxidative burst, and Ca^2+^-dependent distant signaling. These responses also include the regulation of plasmodesmata-mediated cargo movement between adjacent cells via callose deposition to suppress pathogen spread, phytohormone production to mediate crosstalk between various signaling components, systemic acquired resistance (SAR) to activate defense signaling in distant tissues, and programmed cell death accompanied by a hypersensitive response at infected regions to restrict pathogens and prevent their spread to non-infected areas [48,49]. Overall, ETI and PTI responses are activated through complex and overlapping mechanisms. However, it remains unclear whether these mechanisms follow a specific order and how or where each mechanism triggers particular layers of the plant immune system.

Additionally, plants have evolved extra layers of immunity to defend against specific biotic agents. These include the strict regulation of stomatal opening and closure to prevent water loss or biotic agent entry, physical barriers such as leaf surface wax, cell wall lignification, thorns, cuticles, and trichomes, as well as the secretion of antimicrobial peptides (AMPs) and toxic specialized metabolites [49,58,59]. Antimicrobial peptides are short proteins comprising 15–150 amino acids with anti-oomycete, antifungal, antiviral, and antibacterial activities. They are found in all eukaryotes. Recently, a significant number of AMPs derived from plants and animals and artificially designed AMPs have been documented [49,58]. In some cases, plant resistance reduces pathogenic infection rather than providing qualitative and almost complete inhibition, a phenomenon termed quantitative disease resistance (QDR). Multiple genetic loci control QDR, termed quantitative trait loci (QTLs), are associated directly or indirectly with plant immunity and disease resistance [60]. RNA silencing is one of the most effective plant defense mechanisms, playing a central role in combating viral infections. Non-viral pathogens also activate this machinery, highlighting its importance beyond antiviral responses.

Additionally, most eukaryotes employ RNA silencing as a gene regulation method to coordinate growth and developmental processes [61]. Significant examples include RNA interference (RNAi), facilitated by non-coding RNA molecules like small interfering RNAs (siRNAs) or microRNAs (miRNAs). Consequently, pathogen-triggered RNA silencing can disrupt other endogenous processes. In the case of insect pest attacks, plants possess a complex and dynamic defense system, which includes structural barriers, toxic chemicals, and the ability to attract natural enemies of pests. These defense mechanisms can be either constitutive (always present) or inducible (activated after herbivore damage) [59]. Constitutive barriers include thick cell walls, trichomes, and the production of secondary metabolites that deter herbivores. Inducible defenses in plants are triggered when herbivores attack, initiating a series of molecular events. First, plants recognize herbivore-associated molecular patterns (HAMPs) through specific receptors, which then trigger a defense response (Figure 3), leading to a cascade of signaling events involving plant hormones such as JA, SA, and ethylene [58,59]. These signals activate the expression of various defense-related genes, which encode proteins for direct defenses, such as proteinase inhibitors and secondary metabolites, as well as indirect defenses like volatile organic compounds. The produced compounds deter herbivores by inhibiting their digestive enzymes, reducing palatability, or disrupting their digestion and metabolism. Leaf-eating insect pests, such as caterpillars, beetles, and grasshoppers, cause significant mechanical damage to plant tissues when they feed on leaves, reducing the photosynthetic capacity of plants. This also creates entry points for pathogens, compounding the plant stress. Overall, plants with depleted resources and weakened defense mechanisms due to HS or other biotic and abiotic stressors are more susceptible to attacks by biotic agents, owing to shared plant immune mechanisms.

## 5. Plant Responses to Concurrent Heat Stress and Biotic Interactions

Heat stress can affect biotic stressors and influence their interaction with the host plant (Figure 4). Therefore, studying and understanding the effects of combined stress encounters and their interactions concerning plant defense responses is crucial. Recent studies have shown that temperature changes can positively or negatively impact plant immune responses during pathogenic and parasitic infections. As discussed in the previous sections, plants have developed various strategies to cope with individual stresses while minimizing their impact on fitness. However, managing the trade-off between defense and growth during physiological and molecular adjustments to combined stresses presents a significant challenge for plants. Adaptations to one type of stress can make plants more susceptible to another, as seen in the mechanism of stomatal regulation [62]. For example, excessive stomatal opening, which occurs as an early adaptation to HS to cool the leaves, can be exploited by pathogens. Likewise, water loss during combined HS–drought conditions would be harmful. The genetic basis for plants being more susceptible or tolerant to specific pathogens under HS is not well understood [63].

Here, we summarized over 80 studies examining the interactions between HS and specific biotic stresses (Table 1). These reports include model plants and crops such as *Arabidopsis*, tomato, wheat, soybean, rice, chili pepper, fruit crops, and others. In the following sections, we emphasize the role of HS and individual phytopathogenic and beneficial biotic agents, along with their effects on plant physiology and immune responses, based on the findings reported in current literature.

### 5.1. Plant–Heat Stress–Bacterial Interactions

The interaction between HS and pathogenic or non-pathogenic bacteria is critical for plant health and climate resilience. Thus, studies about dissecting the intricate effects of HS on bacteria and plant–bacterial interactions are vital. High temperatures can reshape the bacterial communities surrounding plant roots (rhizosphere) and influence the types of chemicals plants release from their roots (root exudation) [137]. Elevated temperatures can increase or decrease plant immune responses to bacterial infections (ETI and PTI). For instance, *Arabidopsis* plants infected with *Pseudomonas syringae* pv. Tomato DC3000 (*PsPto*) causing bacterial speck under controlled conditions exhibited ISR activation and HR suppression simultaneously when subjected to a gradual temperature increase from 22 to 28 °C [64]. During this HS–bacterial–plant interaction, the expression of multiple plant immunity-related genes (*EDS1*, *PAD4*, *RPS2*, *RPS4*, and *RPM1*) was modulated. However, temperature shifts to 42 °C for 1 h and 37 °C for 2 h reduced plant tolerance mediated by *FLS2* during *PsPto* infection [63]. Furthermore, elevated temperatures from 22 to 28 °C and 30 °C reduced *PsPto* tolerance due to HR suppression involving the *ZAR1* and *RPS2* genes [64] and *SR1/CAMTA3*-mediated compromised stomata closing [67].

In *Ralstonia solanacearum* infection, *Arabidopsis* showed reduced tolerance at 27 °C and 30 °C, mediated by the *RPS4/RRS1-R* genes [68]. Similarly, higher temperatures (29 °C and 35 °C) rendered resistance less effective against *Xanthomonas oryzae* pv. oryzae in rice, mediated by QDR-related genes [69]. Conversely, lines with the *Xa7* QTL showed tolerance to *X. oryzae* pv. oryzae infection under both controlled and field conditions, highlighting the role of specific QDR genes like *Xa7* in conferring resistance [69,70]. Furthermore, a recent study demonstrated that the line with *Xa4*+*Xa7* exhibited tolerance, indicating the synergistic effect of these genes in enhancing immunity [71]. Also, temperature variations influenced the rice responses to *X. oryzae* at different developmental stages [72]. 

Other studies involving various plant species such as tomato [73,74,75], pepper [74,76,77,78], and tobacco [74] have also highlighted the intricate relationship between HS (elevated temperatures), bacterial pathogen resistance, and plant immune responses (Table 1). For instance, Yang et al. [74] found that *Ralstonia solanacearum* infection in pepper triggers SA and JA signaling pathways at different stages under average temperature. Still, these responses were hindered under HS, making plants susceptible to pathogens. Exogenous trans-zeatin (cytokinin) treatment enhanced disease resistance against *R. solanacearum* under HS conditions in Solanaceae species (tobacco, pepper, and tomato), likely due to cytokinin-mediated induction of chromatin remodeling and modulation of genes associated with defense mechanisms. 

One overlooked but concerning group of bacterial phytopathogens includes phytoplasmas, infecting insects and plants [138]. Phytoplasmas are mycoplasma-like bacteria restricted to the phloem and transmitted by insects, affecting various plants such as ornamentals, weeds, fruit trees, vegetables, and other cultivated crops [139]. The impact of phytoplasma infections has been documented in different crops, including sesame, maize, tomato, chickpea, vineyards, and chrysanthemum blooms [140,141,142,143,144,145]. In the past decade, hundreds of new phytoplasma species have been reported in warmer regions, and rising temperatures are expected to significantly affect the distribution of phytoplasma-transmitting insect vectors [146]. High temperatures directly affect plants and modulate various aspects of the phytoplasma disease cycle. These include replication and movement within infected plants and the interactions between plants, insects, and phytoplasmas (host–vector–pathogen). For instance, faster phytoplasma multiplication in the plant hosts exposed to elevated temperatures increases the chances of vector transmission during feeding by leafhoppers [147,148,149]. 

Overall, HS is a pivotal contributor to the severity and dissemination of plant–bacterial interactions, warranting detailed investigations. Gaining molecular insights into the intricate interplay between HS, plant–bacterial interactions, and the underlying molecular mechanisms of plant immunity will help design novel approaches for crop protection. Ultimately, understanding these complex interactions will facilitate the development of climate-resilient crops.

### 5.2. Plant–Heat Stress–Fungal Interactions

Heat stress significantly impacted plant–fungi interactions, influencing both the eukaryotic interaction partners in aboveground and belowground ecosystems. Several studies have examined how HS affects fungal diseases such as blast, sheath blight, dry root rot, leaf rust, and root rot in different crops, including rice, chickpea, coffee, American chestnut, common wheat, durum wheat, oat, and barley (see Table 1). Understanding the effects of HS is crucial for developing effective strategies to minimize agricultural losses. This section explores the different responses observed in various plant–fungi interactions under HS conditions.

One notable example is rice, where the blast disease caused by *Magnaporthe oryzae* exhibits diverse responses [79,80,81]. Different QTLs associated with QDR respond differently to temperatures ranging from 38 to 35 °C. Some investigated QTLs were found to decrease resistance, while others, such as *Pi54*, demonstrated enhanced tolerance through QDR and ETI mechanisms [79,80]. In the case of sheath blight in rice, elevated temperature studies conducted under semi-field conditions showed reduced tolerance [82]. These variable plant responses underline the complexity of rice–fungus interactions under HS. In chickpea cultivation, *Macrophomina phaseolina* can worsen the threat of dry root rot under HS conditions [84]. Vital defense proteins such as Endochitinase and CHI-III exhibited decreased expression, compromising the plant’s ability to fend off fungal attacks [83].

Similarly, coffee plants facing leaf rust caused by *Hemileia vastatrix* exhibited reduced tolerance under HS, albeit shading and high nitrogen levels can partially alleviate the adverse effects of HS [86]. American chestnut trees afflicted by root rot (*Phytophthora cinnamomi*) experienced disrupted plant-defense responses, underscoring the critical consequences of HS on plant health [87]. Rising soil temperatures due to climate change-induced HS were reported to decrease plant tolerance to the fungal pathogen, as evidenced by models predicting reduced biomass and forest landscape changes, highlighting the urgent need for adaptive strategies to safeguard chestnut plantations.

Wheat, a staple crop worldwide, is susceptible to various fungal diseases that respond differently to HS. Some *R* genes that provide resistance against pathogens, like *Puccinia graminis* f. sp. *tritici* (stem rust), were less effective under elevated temperatures. However, some studies have shown that certain QTLs and *R* genes can maintain or even enhance disease resistance (see Table 1). These include resistance against *Blumeria graminis* f. sp. *tritici*, *Puccinia recondita* (causing leaf rust), and stripe rust-causing *Puccinia striiformis* f. sp. *tritici*. These studies highlight the complex and nuanced interaction between HS and fungal diseases in wheat, emphasizing the importance of tailored disease management strategies. Similar complex responses were observed in oats [104] and barley [105,106,107,108,109], where fungal diseases pose significant challenges under HS conditions. Some cultivars exhibited diminished resistance, while others maintained stable or enhanced tolerance. 

Symbiotic fungal associations, such as arbuscular mycorrhizal fungi (AMF), can alleviate heat and other abiotic stresses in plants [150]. However, the impact of combined heat and biotic stressors in the presence of AMF associations requires further investigation. In summary, the effects of HS on plant–fungus interactions have significant implications for diverse crops, challenging traditional approaches to disease management in the era of climate change. Heat stress disrupts the delicate balance between plant–fungal stress interactions, leading to varied outcomes, including reduced tolerance or stable resistance, depending on the specific pathogen and host interaction. Therefore, it is crucial to consider the effects of HS on plant–fungus interactions to develop effective and sustainable agricultural practices.

### 5.3. Plant–Heat Stress–Viral Interactions

HS has significantly impacts virus and viroid infections in various plant species, influencing plant immunity, the severity of disease symptoms, and stress-response mechanisms. Elevated temperatures can accelerate virus transmission through insect vectors, increase viral load, replication, or movement inside infected tissues, and enhance symptom severity across multiple plant–virus interactions [151,152,153,154,155]. Furthermore, the defense pathways induced by HS and viral infections overlap with each other and, thus, pose a complex challenge to the plant system when dealing with concurrent HS and viral infections (summarized in Table 1). Combinatorial HS–viral stress weakens plant immunity that seems to stem from the potential of HS to inhibit the activation of defense genes, HSP production, and altered RNA silencing and SA pathways, which are crucial components of the plant immune system [156,157]. While some HSPs have been found to aid viruses in replication and spread throughout the plant [155,158], others appear to bolster the plant’s antiviral defenses [159,160]. 

Understanding the intricate responses of plants to viral infections under HS conditions provides valuable insights into plant–HS–viral interactions. For instance, tobacco plants exposed to HS during infection with the *Tobacco mosaic virus* exhibit necrosis and ETI-mediated HR suppression [114]. Similarly, potato plants infected with *Potato virus Y* (*RSV*) under growth chamber conditions experienced HR suppression response when exposed to elevated temperatures [115]. In rice, exposure to the *Rice stripe virus* in cell suspension cultures under gradual temperature increase resulted in the suppression of viral replication and symptom development [161]. Furthermore, pepper plants infected with *Paprika mild mottle virus* displayed similar trends, with HR being less effective and systemic infection prevalent [123]. In contrast, maintenance of stable viral resistance was also observed in some scenarios when exposed to variable HS conditions. For example, rice plants infected with *RSV* retained stable resistance conferred by the *Stvb-I* gene [117]. Likewise, tomato plants infected with *Tomato yellow leaf curl virus* exhibited enhanced expression of several HSPs, contributing to stable viral resistance. However, impairment of the HS-responsive HSF2A protein, caused by interactions with viral components, suppressed the activation of downstream HS-responsive genes and compromised the HS response [118].

Tobacco plants also showed variable responses depending on the virus and the defense pathways activated during HS tests. For instance, resistance mediated by the *Rx* gene against *Potato virus X* was suppressed under HS conditions, reducing HR response [64]. Transcriptional studies of *PR* and *HSP* genes in potato plants subjected to both *Potato virus Y* infection and HS highlighted that PVY infection and high-temperature stress shared specific regulatory mechanisms with HS, directly affecting the stability of *R* genes-encoding proteins and leading to the HR inhibition during viral infection [114]. Wheat crops exhibited differential resistance to *Wheat streak mosaic virus* depending on temperature ranges and the presence of specific *R* genes and related pathways in different cultivars [121,122,161]. In pepper plants infected with *Tobacco mild green mosaic virus*, the HR response was suppressed when mediated by *L1–L2*, but *L1a*-mediated resistance showed stable resistance even at higher temperatures [123]. Common bean plants exhibited a mixed reaction to HS when infected with *Bean pod mottle virus* [124]. Susceptible lines showed severe symptoms under HS, while resistant lines demonstrated stable inhibition and restricted viral spread. In the case of *Capsicum chlorosis virus*, HS initially promoted viral replication in Capsicum. Still, at a later stage of infection, systemic recovery occurred in newly emerging leaves resulting from the activation of RNA silencing machinery [119]. 

Taken together, no plant–viral pathosystem can serve as a universal model to investigate the effects of HS on plant–virus interactions, pressing the need for multifactorial and comprehensive studies to map governing factors. Also, plant varieties with natural resistance against viruses under HS can be used in breeding programs, or resistant cultivars can be developed through molecular breeding and biotechnology approaches.

### 5.4. Plant–Heat Stress–Nematode Interactions

Plant–parasitic nematodes pose a more significant threat to plant health when HS situations weaken plant defenses. For instance, HS compromises soybean resistance to the soybean cyst nematode [162,163]. This vulnerability is also observed in tomatoes and watermelons when facing root-knot nematodes under HS conditions [164]. Constant efforts have been made to unravel the complex mechanisms behind how HS and other stress agents affect the JA signaling pathway in plants, which is crucial for warding off nematodes [165]. In soybeans, HS disrupts resistance to nematodes and throws their hormonal balance into disarray [166,167]. 

Delving into plant–nematode interactions sheds light on the mechanisms governing resistance to these pervasive pests. In wheat and other cereal crops, the cereal cyst nematode *Heterodera avenae* is an important soil-derived pathogen investigated to gain molecular insights into resistance mechanisms [168,169,170]. However, there is a lack of literature on the plant molecular responses to combined HS and cyst nematode infection, indicating the need for further research. Tomato plants challenged by *Meloidogyne incognita* under greenhouse conditions inhibited ETI-mediated resistance [125]. However, pepper plants facing various *Meloidogyne* species under HS displayed stable HS resistance with ETI activation [129]. Most current practices involve toxic and environmentally damaging chemical nematicides to control nematodes. Future studies will continue to explore the interactions between HS and plant–nematode dynamics, providing insights that will influence ecofriendly agricultural practices and crop management strategies.

### 5.5. Plant–Heat Stress–Insect Interactions

Plant-interacting insects, including pests and pollinators, are highly susceptible to HS, which affects their behavior, physiology, and performance [171,172,173]. Distinct insect species also act as vectors, transmitting pathogenic bacteria, viruses, and phytoplasmas. Heat stress can cause insect mortality and reduce agricultural product quality [174,175]. Elevated temperatures challenge both insects and their symbiotic partners [176,177]. Insects may exhibit altered behavior and decreased performance [178,179]. Plant–insect interactions are adversely affected during the HS encounter of plants via altered metabolome, reduced photosynthesis, and compromised plant growth [171,180,181,182]. In addition, elevated temperatures are a critical factor in changing insect host preferences [171,172]. Interspecies competition between two insects was also impacted by HS, as evident in wheat aphid species *Rhopalosiphum padi* and *Sitobion miscanthi* [183]. Overall, insect and plant life cycles are prone to HS while interacting with each other or during independent phases of their lifecycles. Further research will provide valuable insights into insect and plant responses to changing environments, and understanding these dynamics is pivotal for effective pest control and ecological adaptation to climate shifts.

In tomato plants, exposure to *Helicoverpa armigera* on an artificial diet under gradual temperature increase in growth chamber conditions induced robust resistance against caterpillar feeding [184]. Similarly, in wheat and soybean, aphid infestation by *Rhopalosiphum padi* and *Aphis glycines*, respectively, on an artificial diet under gradual temperature increase in growth chamber conditions led to reduced aphid reproduction and population growth [136,185,186,187,188,189]. Moreover, in rice, exposure to *Chilo suppressalis* on an artificial diet under gradually increasing temperatures in growth chamber conditions induced resistance against insect feeding and reduced larval growth [190,191]. In nature, insects feed on living plants, and the rearing method can significantly affect their development and reproduction. Most studies on HS–insect interactions use detached leaves for rearing, which may offer lower nutritional quality than living plants [136]. This discrepancy can lead to different conclusions. Therefore, conducting research with insects reared on living plants would likely yield more accurate results, particularly when evaluating their performance under HS conditions.

Insects are crucial as pollinators, significantly contributing to plant reproduction and crop yields. The interaction between plants and pollinating insects is a brief but vital engagement, where plants provide food resources to the insects in exchange for essential pollination services [192]. However, their ephemeral (short-term) interactions with plants, especially during HS events, require thorough research. Understanding how HS affects these interactions is essential for developing strategies to mitigate potential negative impacts on pollination and crop productivity. As HS becomes more frequent, studying the effects of HS on pollinator behavior and efficiency will be increasingly important for ensuring sustainable agricultural practices and food security. Heat stress can alter the pollinator dependency of crops, indicating that insect-mediated pollination might become increasingly crucial for crop production as the likelihood of heat waves rises. A decrease in plant and flower visits by pollinating insects during HS limits pollen dispersal, ultimately negatively impacting crop yields. This effect has been observed in interactions between buff-tailed bumblebees (*Bombus terrestris*) and faba bean (*Vicia faba* L.) [193], as well as common eastern bumblebees (*Bombus impatiens* Cresson) and canola (*Brassica napus*) [194]. 

In nature, HS and the gradual cooling process led to fluctuating temperatures that significantly affect specific life parameters of all the insects and host plants compared to constant temperatures [195]. Overall, HS negatively impacts pollinators and pollination services [196]. Therefore, it is essential to focus on understanding the potential behavioral and physiological effects of thermal stress on pollinator insects, both independently and in conjunction with interacting plant species.

### 5.6. Plant–Heat Stress–Weed/Parasitic Plant Interactions

Plant–weed interactions, including parasitic plants, can significantly impact crop productivity, weed management, and overall agricultural sustainability. Typically, weedy species are more adaptable and resilient in challenging climates, potentially giving them a competitive advantage over crops. The simultaneous presence of HS and weed competition significantly influences plant growth, productivity, and quality, especially in crops like rice [197,198]. Heat stress also affects the interaction of plants with invasive weed-associated bacteria. For instance, invasive weed-associated bacteria enhance wheat’s tolerance to HS [199]. Crops facing both HS and weed competition can experience changes in the growth and reproductive abilities of certain species, as observed in *Solanum nigrum* [200]. Recent studies have also reported the harmful effects of simultaneous HS and weed competition on maize crops [201]. Temperature can impact the competitive ability of different weed species. Heat stress and weed competition have been found to affect growth, physiological traits, and yield in cucumbers [202]. Heat stress can also influence the effectiveness of herbicides in weed control for maize [203]. Heat stress has been observed to influence growth and development in various weed species [204]. Moreover, HS and weed interference can impact the yield components of durum wheat [205]. Australian annual and perennial grass species have shown varied responses to HS [206]. The impact of HS and drought stress on the growth and yield of *Amaranthus* species has also been investigated [207].

Parasitic plants have a longer lifespan than microbial invaders. They depend on host plants for food and have an obligatory nature, which requires close interaction with their hosts. Many aspects of host–parasitic plant interactions and the competition between main crops and weedy species, especially in climate change and HS, are still unknown. Future research will reveal these dynamic interactions among plants. In conclusion, the interactions between crop plants and weedy species during HS are complex and multifaceted, involving competition for water, nutrients, and light resources. They also include physiological adaptations such as stress tolerance and allelopathy and management challenges such as herbicide effectiveness. Addressing these interactions requires integrated approaches that consider both the biological and ecological aspects of crops and weeds.

## 6. Perspectives on Dealing with Combined Heat and Biotic Stresses in Agriculture 

Recent research has focused on the molecular interactions between high temperatures and biotic plant stresses, highlighting the importance of improving plant health to thrive under stress combinations to enhance crop productivity and sustainability in changing environments. Moreover, the importance of plant growth-promoting (PGP) microbes in alleviating stressful events [208] highlights the potential of beneficial microbes in improving plant stress resilience. In this section, we explore perspectives based on the following points: the potential use of microbiomes for multiple-stress alleviation, constructing accurate regulatory networks for plant responses to combined HS and biotic stresses, creating reliable stress prediction models, priming plants to activate defense responses before stress occurs, and developing multi-stress-resilient crop varieties alongside innovative agricultural practices.

### 6.1. Potential Use of Plant Microbiome

The microbiome supports plant health in a changing climate [209,210,211]. Microbes and their byproducts play a pivotal role in fortifying plants against biotic and abiotic stressors, enhancing their resilience [212]. These interactions occur through intricate relationships with their host plants, mediated by the plant microbiome [213]. Plants often confront harsh environmental conditions that disrupt their biological processes and developmental pathways [214]. Studies on beneficial microbes that enhance plant tolerance to HS and mitigate biotic stress effects are summarized in Table 2 and Table 3, respectively. Figure 5 depicts the HS effects on host immune responses and the potential of the microbiome to fight against combined stresses. It would be interesting to find out how these beneficial microbes perform and whether they can aid in plant defense during encounters with combined HS and biotic stress circumstances.

Microorganisms, rich in beneficial metabolites, assist plants in coping with these stresses, fostering a diverse ecosystem characterized by mutualistic interactions [245]. Extensive research underscores the pivotal role of the microbiome in shaping plant responses to environmental adversities, particularly abiotic stresses like drought and HS [246,247]. Both drought and HS (elevated temperatures) significantly influence the composition and abundance of root and soil microbiomes [248]. Induced systemic resistance is vital to the interactions between plants and their beneficial microbiomes, priming the whole plant system to fend off phytopathogens and pests more effectively [249,250]. ISR primes the entire plant system to fend more effectively against phytopathogens and pests [251]. 

Microbiome engineering enhances plant functions, including resilience to biotic and abiotic stresses, overall plant fitness, and productivity [252]. By manipulating plant microbial communities, microbiome engineering holds promise for boosting crop resilience and long-term agricultural productivity [253,254]. Recent reports suggest the effectiveness of microbial agents in mitigating various stresses, addressing both biotic and abiotic threats [255,256]. However, climate change-driven environmental stresses also disrupt the plant microbiome functioning, impacting plant–microbe interactions and stress responses. Thus, understanding plant–microbiome communication is critical to unlocking the potential of beneficial bacteria in combined stress management [257]. By harnessing the microbiome’s role in plant resilience, microbiome engineering offers innovative pathways for balanced agriculture and environment [258]. Harnessing the role of microbiomes in plant resilience through microbiome engineering provides creative ways for environmental protection and sustainable agriculture. Given the evolving environmental landscape, integrating microbiome expertise into agricultural methods will help enhance plant efficiency and ensure food and nutrient stability (See Figure 5).

**Table 3 plants-13-02022-t003:** Studies investigating the potential use of beneficial microbes to manage phytopathogenic infections and insect herbivore attacks for sustainable plant health management. ISR, induced systemic resistance; SAR, systemic acquired resistance; SM, secondary metabolites.

Plant	Plant-Beneficial Microbes	Pathogen (Disease)	Pathogen/Lifestyle	Details	Ref.
*Arabidopsis*	*Bacillus subtilis* PTA-CT2, *Pseudomonas fluorescens* PTA-271	*Pseudomonas syringae* pv. *tomato* DC3000 (*Psto*)	Bacterial hemibiotroph	Root-drench, priming effect on plant immunity, ISR	[259,260]
		*Botrytis cinerea* (Grey mold)	Fungal necrotroph		
	*Bacillus proteolyticus* OSUB18	*Psto*	Bacterial hemibiotroph	Root-drench, priming effect on plant immunity, ISR	[261]
		*Botrytis cinerea*	Fungal necrotroph		
Soybean	*Trichoderma viride* GT-8, T. *reesei* GT-31, *T. longibrachiatum* GT-32	*Macrophomina phaseolina* (Charcoal rot), *Sclerotinia sclerotiorum* (White mold), *Fusarium* sp. (Seedling damping-off)	Fungal necrotroph	Seed treatment, plant growth promotion (PGP), competition	[262]
	*Trichoderma harzianum*, *T. asperellum* T00	*Pratylenchus brachyurus* (Root lesion)	Nematode	Seed treatment, production of SM, lytic enzymes	[263,264]
Dragon fruit	*T. viride*, *T. asperellum*, *T. harzianum*	*Neoscytalidium dimidiate* (Brown spot)	Fungal	Added to base soil of tree, competition, ISR, antibiosis synthesis	[265]
Kale	*Pseudomonas fluorescens* SP007S	*Pectobacterium carotovorum* (Soft rot disease)	Bacterial necrotroph	Soil treatment, SAR, phenol compounds, PGP	[266]
Olive	*Bacillus* sp., *Pseudomonas fluorescens* (alone or combined)	*Pseudomonas savastanoi* (Knot disease)	Bacterial necrotroph	Foliar spray, SM, PGP	[267]
Chili pepper	*Bacillus thuringiensis* MW740161.1, *Pseudomonas fluorescens*, *Bacillus subtilis*	*Leveillula Taurica* (Powdery mildew)	Fungal	Foliar spray, production of SM, peroxidases, ISR	[268]
Sorghum	*Trichoderma viride*	*Dickeyadadantii* (Stalk rot)	Bacterial	Seed treatment, production of defense enzymes, PGP	[269]
Potato	*Trichoderma harzianum*, *Pseudomonas fluorescens*	*Alternaria solani* (Early Blight)	Fungal	Tuber treatment, PGP	[270]
Cucumber	*Trichoderma harzianum*	*Fusarium oxysporum*	Fungal	Seed treatment, production of defense enzymes, PGP	[271]
Tomato	*Trichoderma asperellum*	*Agroathelia rolfsii* (Collar rot)	Fungal necrotroph	Root-drench, production of hydrolytic enzymes, SM	[272]
	*Trichoderma longibrachiatum*, *T. atroviride*, *T. harzianum*	*Alternaria solani* (Early Blight)	Fungal	Foliar spray, production of SM, antioxidant enzymes, PGP	[273]
	*Trichoderma asperellum*, *T. harzianum*, *Bacillus subtilis Verticillium lecanii*, *Metarhizium anisopliae*,	*Meloidogyne* spp. (Root-knot nematode)	Nematode	Cavity chamber method, production of lytic enzymes	[274]
Cucumber, tomato	*Pseudomonas protegens* 1B1, *P. clororaphis* 48G9, *P. brassicacearum* 93G8	*Agrobacterium rhizogenes* (Hairy root disease)	Bacterial	Root dip method	[275]
Wheat	*Beauveria bassiana*, *Metarhizium anisopliae*	*Rhopalosiphum rufiabdominalis* (Aphid)	Insect	Foliar spray, penetration of hyphae through the insect cuticle	[276]
	*Beauveria bassiana* (Balsamo) Vuillemin	*Tenebrio molitor* (Mealworm), *Monochamus alternatus* (Japanese pine sawyer), *Allomyrina dichotoma* (Japanese rhinoceros beetle)	Insects	Syringe application, production of toxins	[277]
Sugarcane	*Metarhizium anisopliae*	Termite (*Odontotermes obesus* (Ramb.), *Microtermes obesi* (Holgren)	Insect	Soil drenching, muscle contraction, flaccid paralysis	[278]
	*M. anisopliae*	*Holotrichia serrata* (White grub)	Insect	Soil application, protease production, flaccid paralysis	[279]
Walnut	*Lecanicillium lecanii*, *Metarhizium anisopliae sensu lato*, *Beauveria bassiana*	*Myllocerus fotedari* Ahmad (Weevil)	Insect	Foliar spray, production of toxin and flaccid paralysis	[280]
Orange and mandarin	*Bacillus thuringiensis*, *Metarhizium anisopliae*, *Trichoderma harzianum*	*Monacha obstructa* (Land snail)	Animal pest	Leaf dipping, poisonous baiting, and foliar spray, hyper parasitism	[281]

Plants harbor a diverse community of microbes, collectively called the phyto-microbiome, including bacteria, fungi, viruses, and archaea [282]. Symbiotic microbial communities promote plant growth, development, and resilience against environmental challenges [283,284]. The plant microbiome facilitates nutrient uptake, synthesizes beneficial hormones, and protects against detrimental pathogens [285]. Certain microbial inhabitants even produce stress-responsive compounds that directly stimulate plant growth under harsh environmental conditions [286]. This intricate web of interactions empowers plants to endure and thrive under challenging circumstances such as drought, HS, or salinity [287]. Beyond abiotic stressors, the plant microbiome is pivotal in preventing biotic threats such as insect pests and pathogens [288]. Beneficial microbes residing in both the leaves (phyllosphere) and roots (rhizosphere) activate innate defense mechanisms, priming them to combat invaders [289] and protecting the host plants by antagonizing pathogens using synthesized microbial metabolites [290]. The interaction between innate immunity and the plant microbiota is crucial in determining the ability of plants to defend against pathogenic attacks [291]. Certain microbes further augment this defense by producing antimicrobial compounds, acting as natural biocontrol agents. However, the efficacy of these biocontrol agents depends on various influencing factors, underscoring the intricacies of plant–microbe–pathogen interactions [292].

Heat stress significantly shapes the root and soil microbiomes, influencing plant–microbe interactions [293]. For foliar diseases, direct spraying of microbial agents onto the infected plant parts has demonstrated efficiency for controlling the disease. Comprehending these interactions can pave the way for sustainable strategies in crop enhancement and stress alleviation [294]. Exploring the crosstalk between responses to HS and biotic stresses will assist in unveiling the intricate modulation of the plant immune system [9].

Various microbial products are available on the market as biocontrol agents (pathogen and pest control), biostimulants, and biofertilizers. The adequate and predictable use of these products depends on selecting species and strains that are suited to specific climatic zones and local environmental conditions due to the strong influence of temperature on plant–microbe interactions. For instance, the mesophilic fungus *Trichoderma* spp., known for its multiple biocontrol capabilities against various pathogens and insect pests [132,295,296], faces constant threats from HS and other climate change-related stresses. Empowering the beneficial microbiome with HS-resistant properties through modern genetic approaches can significantly improve their effectiveness under HS conditions. This has been demonstrated in *Trichoderma viride* Tv-1511, overexpressing the HSP protein (TvHSP70) [297]. Molecular analyses showed that the TvHSP70-overexpressing strain exhibited HS resistance and significant improvements in growth, antioxidant capacity, and antifungal activity compared to the wild-type strain. Thus, adopting such advanced techniques can ensure the reliable performance of microbial products for biotic stress management under varying environmental stresses. By elucidating the molecular mechanisms, we can pave the path toward enhancing stress tolerance and fostering sustainable agriculture amidst shifting environmental conditions [298,299]. Diverse microbial resources under heat–bacterial stress conditions would be instrumental in shaping future experiments and outcomes.

### 6.2. Building Precise Regulatory Networks of Plant Responses to Combined Heat and Biotic Stresses

From molecular and physiological perspectives, investigating plant stress tolerance mechanisms is central to modern plant research. Plants use specific and non-specific responses to cope with environmental stresses, utilizing various molecular components (summarized in Figure 2, Figure 3 and Figure 4). However, the traditional focus on individual stress response pathways is not sufficient. Biotic and abiotic stress pathways are interconnected within a broader network, and plants will face challenges from combined stress factors in the near future [300,301]. Studies have revealed a complex network of molecular interactions that enable plants to efficiently deal with specific singular stress while balancing resource distribution for growth and defense.

Moreover, increasing evidence from field, controlled, and molecular experiments indicates that plants counter specific combined stresses in a non-additive way. In some cases, they even elicit opposite responses, resulting in effects that cannot be predicted solely by investigating singular stresses [62]. Therefore, a key goal of future research should be to explore plant response mechanisms involved in imparting multi-stress tolerance, which is crucial given the forecasted climate changes and the emergence of new stress combinations in agriculture. Identifying genetic factors involved in both HS and biotic stress pathways will provide potential targets for genetic manipulation, such as master regulatory genes and specific transcription factor families.

Emerging omics technologies and artificial intelligence (AI)-based applications offer a powerful toolkit to elucidate molecular mechanisms involved in plant stress [302,303]. Genomics approaches like Genome-wide association studies (GWAS) can pinpoint critical genes associated with HS tolerance. At the same time, transcriptomics through RNA sequencing (RNA-seq) reveals how gene expression changes under HS conditions [304]. Proteomics and metabolomics further explore the specific proteins and metabolites produced or modified in response to HS, providing insights into potential targets for manipulation [305,306]. By integrating data generated from various omics approaches through AI tools, researchers can understand the intricate interplay between genetic elements, proteins, and metabolites during combined abiotic stresses [307]. This holistic view will facilitate the identification of vital regulatory points within the HS and biotic stress response pathways, which can be targeted for manipulation to enhance plant resilience. These omics technologies can accelerate breeding programs by identifying specific markers for combined HS–biotic tolerance and uncovering novel mechanisms [308].

### 6.3. Developing Reliable Stress Prediction Models and Priming Plants against Combined HS–Biotic Stresses

Predicting the combined effects of HS and biotic stress on crops is critical to avoiding crop loss. The exposure timing of HS is critical considering the plant’s developmental stage, which can result in higher crop loss or stress acclimation [309]. Sophisticated computational models are emerging as powerful tools to address this challenge of predicting multiple stresses and their severity [310,311,312]. These models integrate diverse data sets encompassing plant physiology, pathogen biology, and future climate projections [312,313]. By analyzing these datasets through AI-based machine learning algorithms, the models can identify patterns and predict the potential impact of HS on plant–biotic interactions in specific regions [314]. This predictive capability can assist in developing targeted stress management strategies [315]. Furthermore, these models can guide breeding programs by identifying existing stress-tolerant varieties and highlighting areas where new cultivars with enhanced resilience are needed. While challenges like data availability and model validation need to be dealt with, the potential benefits of predicting combined HS–biotic stress interactions make this a crucial area of research.

If we can predict stress patterns, we can also prepare plants to fight against them better. Plants possess a remarkable ability known as stress memory and priming, where exposure to mild stress can prime them for a more robust response to a subsequent stressor [316,317,318]. For instance, heat priming has been reported to induce HS tolerance in wheat [319] and seagrass [320]. Pre-exposure to mild HS can prime plants for enhanced defense against future biotic stress. By elucidating the underlying molecular mechanisms of stress memory and priming in combined stress scenarios, this research can pave the way for developing targeted priming techniques to improve plant resilience [321]. Priming SAR through heat treatment is a promising and cost-effective approach to fighting pathogens with minimal environmental impact [113]. Additionally, recent findings suggest that using a combination of warming treatments and exogenous application of chemical inducers can enhance plant immunity by activating multiple defense pathways simultaneously [322]. Exploiting genetic or induced acclimation strategies to changing ambient temperatures in agriculture can improve multi-stress tolerance in crops.

### 6.4. Developing Stress-Resilient Crop Varieties along with Novel Agricultural Practices

One critical aspect of combined stress studies is that experiments must simulate natural field conditions for meaningful progress. Plant responses can vary significantly under different experimental setups, as evident in the summarized reports in Table 1. Presently, available single HS or biotic stress-tolerant crop varieties are ideal candidates to be tested against a range of stress combinations, intending to improve tolerance to combined HS and biotic stresses. The strategy of pyramiding stress-related genes, which confer resistance to various stresses, into elite cultivars would be helpful. Gene pyramiding can be achieved through molecular breeding, transgenic, and genome editing (GE) approaches [323]. A shift in focus is needed in plant stress research, considering that plants often trade-offs between growth and stress tolerance, resulting in yield penalties. Therefore, developing crop varieties with inherently enhanced abilities for higher photosynthesis, growth, and yield would serve the purpose better than merely surviving stresses.

Improving plant resilience against the combined stresses of HS and biotic factors is critical due to the increased incidences amid changing climate conditions [324]. To address this challenge, scientists are exploring various novel strategies using molecular tools and techniques. These strategies aim to enhance plant resilience and alleviate the negative impacts of combined HS and biotic stresses on crop productivity. One such strategy is the development of resilient crop varieties, which involves breeding and screening crop varieties for tolerance to both stressors [325]. Combining better-performing crop varieties with plant-beneficial microbes would enhance plant health and nutrient uptake [326]. Implementing climate-adaptive practices involves deploying farming methods that mitigate the impact of combined stressors [327,328,329,330,331]. 

Biotechnology, including omics approaches and AI-based climate prediction platforms, offers potential solutions for addressing plant stress by developing stress-tolerant crop varieties and providing projections on combined stress scenarios. Heat and disease-resistant crop varieties can be designed using GE tools like clustered regularly interspaced short palindromic repeats (CRISPR)/CRISPR-associated protein (Cas) system (CRISPR/Cas), which can withstand combined HS–biotic threats [3,332]. Also, the joint use of designed varieties and eco-friendly methods will enhance the plant’s resilience to multiple stress scenarios, including targeted and biocontrol replacements, for instance, mechanical trapping approaches, pheromones, antagonistic organisms, and exogenic application of biological molecules [333]. 

One of the most significant advancements in crop improvement is the development of faster and more effective GE techniques compared to traditional plant breeding methods [334]. CRISPR/Cas-based GE has become the most commonly employed tool for designing plants with required traits, such as tolerance to abiotic stresses and diseases [3,323,335,336,337]. For example, soybean plants with the *GmARM* gene knocked out using CRISPR/Cas exhibited tolerance to multiple stresses, such as alkali-salt stress and nematode root rot disease caused by *Phytophthora soya* [338]. Although the use of GE is most effective when the genes or genetic factors (e.g., promoter elements, transcription factors, RNA molecules, and epigenetic modifications) responsible for combined stress tolerance are already known, CRISPR/Cas-based tools have immense potential to identify these targets, which can be achieved through screening and generating mutant libraries for individual genes or entire gene families, as well as through directed evolution studies of desired genetic loci [339,340]. Overall, the above-discussed various approaches offer promising solutions to address the complex challenges presented by combined HS and biotic stresses and will be forefront areas of research in plant sciences.

## 7. Conclusions

The impact of combined HS and biotic stresses poses a significant challenge to plant immunity under changing climates. The current work summarizes different reports emphasizing the importance of understanding the complex relationships between host plant–HS–biotic interactions. Further research is necessary to develop stress-tolerant crop varieties and sustainable agricultural practices. Diverse plant-beneficial microbes and modern tools offer promising solutions to enhance plant immune resilience against biotic stressors while mitigating the HS effects. It is clear that plant processes that regulate responses to multiple stresses are not dependent on a single controlling morphological, physiological, or genetic factor. Integrated regulatory networks determine a specific set of responses to a particular multi-stress combination. Therefore, it is crucial to identify these regulatory networks to successfully develop plant varieties with enduring resistance to biotic stresses in the face of climate change. This identification can be accelerated in the future through high-throughput (omics) methods, AI-based modeling and analyses, GE, and other genetic engineering tools.

## Figures and Tables

**Figure 1 plants-13-02022-f001:**
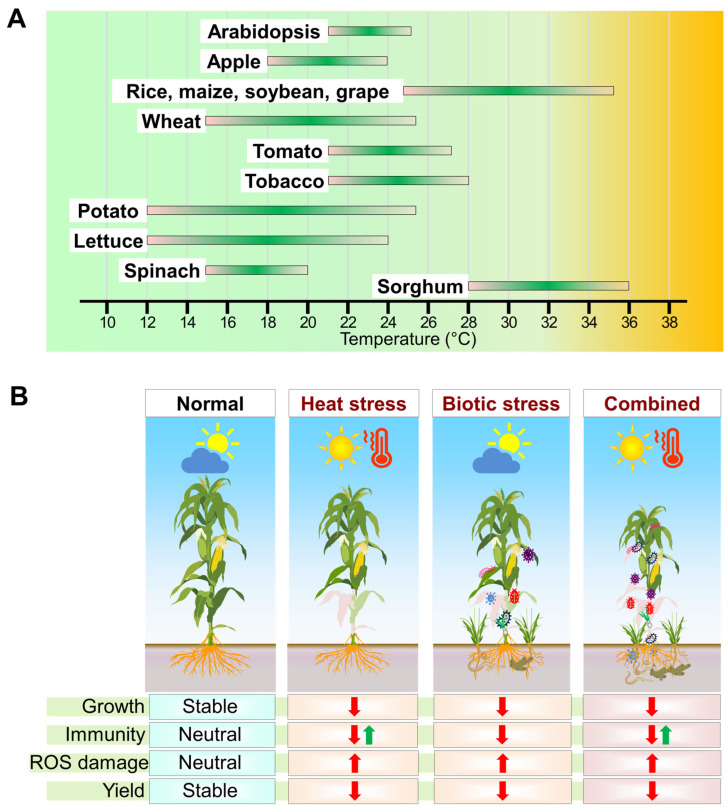
Overview of climate change, heat stress (HS), and biotic stress effects on plant health. This figure depicts an overview of the interplay between climate change, HS, and biotic stress on plant health. Climate change leads to rising temperatures, exacerbating both abiotic and biotic stressors plants face. (**A**) Optimal temperature ranges for the growth and development of major crop species are shown. (**B**) When healthy plants encounter concurrent HS and biotic stresses simultaneously rather than independently, the effect on plant physiology is more severe, as illustrated in this panel. Plant susceptibility results from compromised growth and development, weakened immunity, and reactive oxygen species (ROS)-mediated oxidative damage to cell organelles, ultimately affecting yield and productivity. A green upward arrow indicates a positive impact, while a red downward arrow indicates a negative impact of specific stress on plant traits.

**Figure 2 plants-13-02022-f002:**
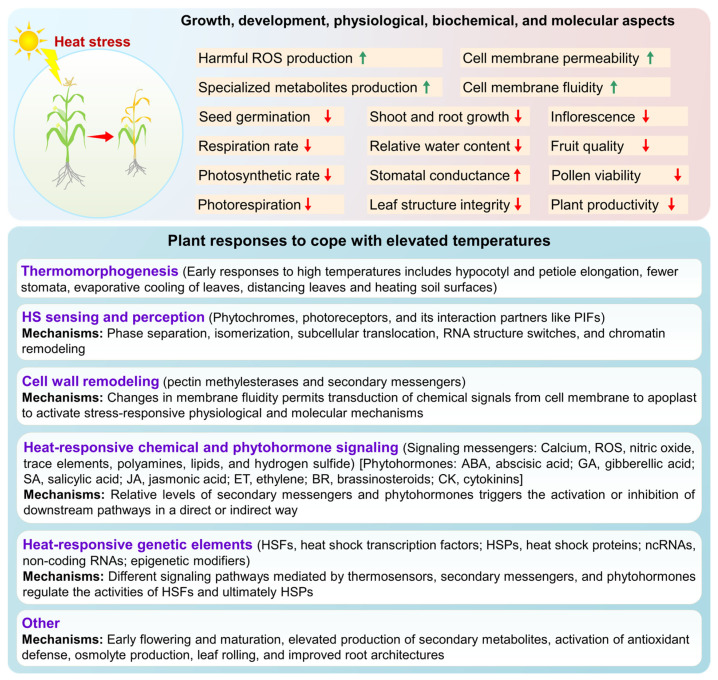
Heat stress (HS)-driven challenges and responses of plants. Plants face various challenges during HS and employ multiple physiological, biochemical, and molecular mechanisms to cope with elevated temperatures. The summarized aspects are discussed in detail in the relevant literature.

**Figure 3 plants-13-02022-f003:**
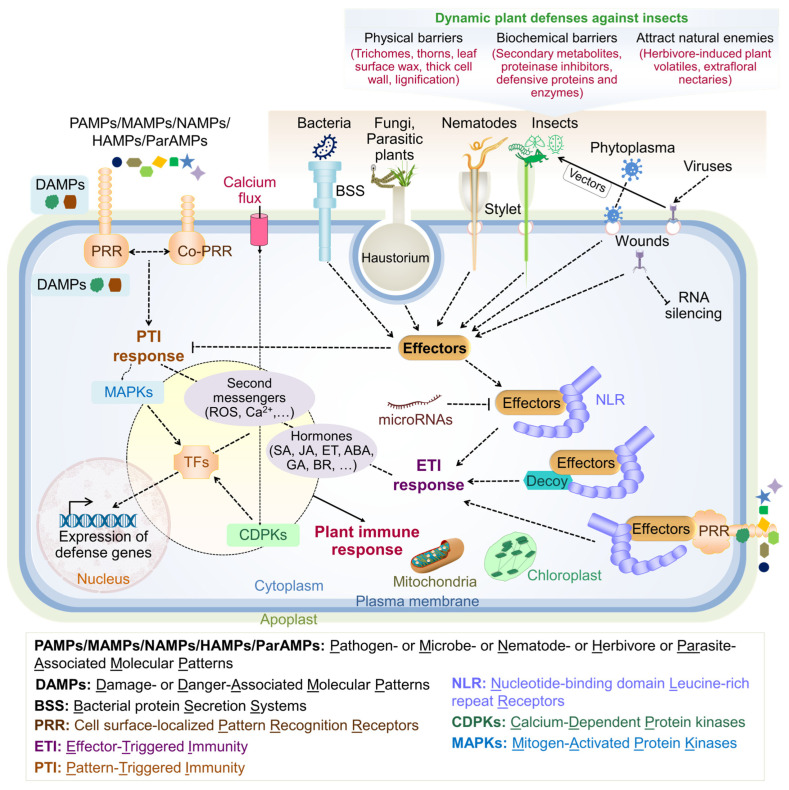
Schematic of plant immune responses to biotic stresses. Biotic agents from different classes (labeled and color-coded) express virulent effectors and specific molecules that are perceived as pathogen-associated molecular patterns (PAMPs) when they colonize plants (shapes are color-coded to represent the particular biotic agents). Insects serve as vectors (carriers) that transmit most viruses and phytoplasmas into host plants. Plant immune aspects are further elaborated in the main text.

**Figure 4 plants-13-02022-f004:**
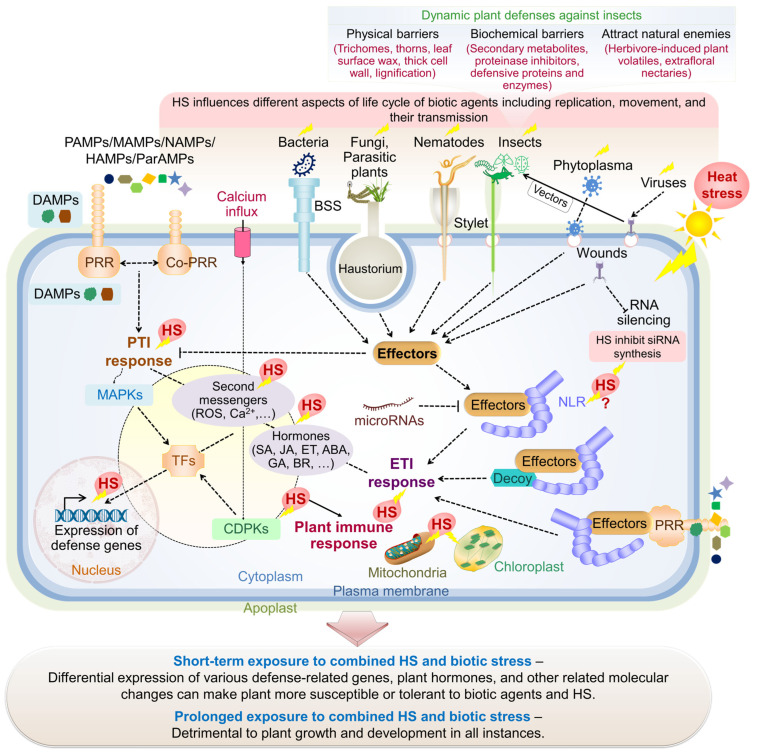
Biochemical and molecular aspects of plant immune responses to biotic stresses overlap with heat stress (HS) responses. The impact of HS on individual biotic agents and their interactions with host plants collectively influence plant defense responses by activating or suppressing overlapping mechanisms. Although some missing or unknown steps exist in the interacting pathways and genes during concurrent HS–biotic conditions, recent studies are beginning to uncover the layout of plant immune responses. The dynamics of plant defense responses are discussed in the main text, along with relevant literature.

**Figure 5 plants-13-02022-f005:**
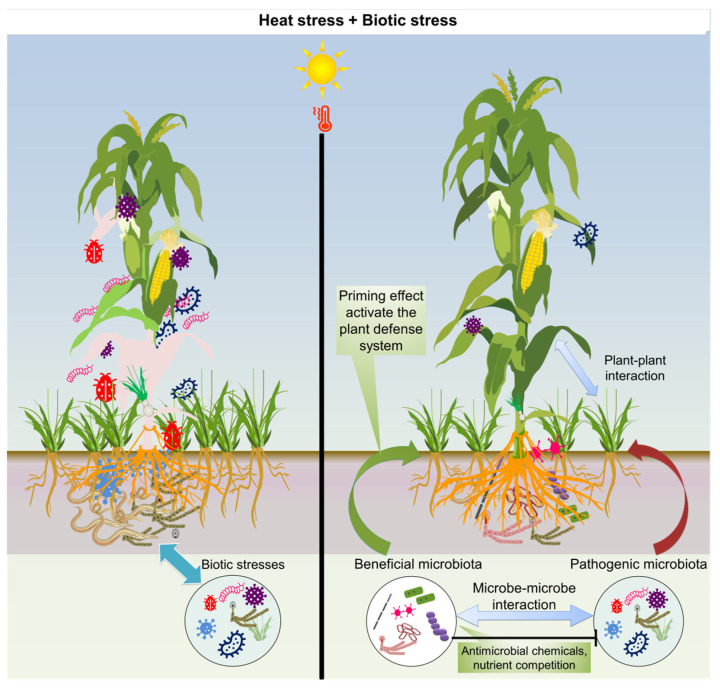
Plant microbiome is critical in enhancing resilience under combined heat stress (HS) and biotic challenges through priming effects on plant immunity. Also, competition for nutrients and antimicrobial chemicals (like antibiotics and toxins) produced by beneficial microbes restricts the growth and spread of pathogenic strains.

**Table 1 plants-13-02022-t001:** Studies investigating plant and significant biotic factor interactions under heat stress (HS) show variable effects on plant health.

Host Plant (Average Temp.)	Biotic Factor	Disease/Nature of Interaction	Set Up	Temp. (°C)	Genes or Other Explored Factors	Effect	Details	Ref.
**Bacteria**								
*Arabidopsis* (22 °C)	*Pseudomonas syringae* pv. *tomato* DC3000 (*PsPto*)	Bacterial speck	C	22, 28	*EDS1*, *PAD4*, *RPS2*, *RPS4*, *RPM1*	(−)	HR suppression (ETI)	[64]
	*PsPto*	Bacterial speck	C	42 (1 h), 37 (2 h)	*FLS2*	(−)	Reduced tolerance (PTI)	[65]
	*PsPto*	Bacterial speck	C	28, 30	*ZAR1*, *RPS2*	(−)	HR suppression (ETI)	[66]
	*PsPto*	Bacterial speck	C	28, 30	*SR1/CAMTA3*	(−)	Reduced tolerance through compromised stomata closing (apoplastic immunity)	[67]
	*Ralstonia solanacearum* GMI1000	Bacterial wilt	C	27, 30	*RPS4/RRS1-R*	(−)	Reduced tolerance (ETI)	[68]
					*SSL4-SSL5*	(+)	Stable HS resistance	
Rice (21–28 °C)	*Xanthomonas oryzae* pv. *oryzae*	Bacterial blight	C	29, 35	*Xa3*, *Xa4*, *Xa5*, *Xa10*	(−)	Resistance less effective (QDR)	[69]
			C, F	29–35 (C) 29–33 (F)	*Xa7*	(+)	Lines with *Xa7* showed tolerance (ETI)	[69,70]
	*X. oryzae* pv. *oryzae*	Bacterial blight	C	29, 35	*Xa4*, *Xa7*	(+)	Line with *Xa4* + *Xa7* showed tolerance (ETI)	[71]
	*X. oryzae* pv. *oryzae*	Bacterial blight	C, F	≥35, 42 (60 h)	*SGS3a/b*, *ARF3a/b*, *ARF3la/lb*	(±)	*SGS3a/b*: (+HS) and (−Pathogen resistance) *ARF3a/b*, *ARF3la/lb*: (−HS) and (+Pathogen resistance)	[72]
Tomato (21–27 °C)	*Ralstonia solanacearum*	Bacterial wilt	C	24, 32	Polygenic resistance	(−)	Resistance inhibition	[73]
	*R. solanacearum*	Bacterial wilt	C	28, 34, 37 (high humidity >80%)	*ITP5*, *trans*-zeatin	(+)	Cytokinin-mediated resistance (ETI, PTI)	[74]
	*PsPto*	Bacterial speck	C	26, 31	Increased ABA, JA, spermine accumulation	(+)	Showed tolerance under mild HS	[75]
Pepper (28 °C)	*R. solanacearum*	Bacterial wilt	C	28, 37	*AGL8*, *SWC4*	(+)	Stable HS resistance (ETI, PTI)	[76]
	*R. solanacearum*	Bacterial wilt	C	28, 37 (high humidity >80%)	*ITP5*, *trans*-zeatin	(+)	Cytokinin-mediated resistance (ETI, PTI)	[74]
	*R. solanacearum*	Bacterial wilt	C	28, 37 (high humidity >80%)	*KAN3*, *HSF8*	(+)	KAN3-HSF8 form a complex to confer immunity and HS tolerance (ETI, PTI)	[77]
	*R. solanacearum*	Bacterial wilt	C	28, 37 (high humidity >80%)	*MLO1*, *PUB21*	(+)	MLO1-PUB21 form a complex and function distinctly in a temperature-dependent manner (stable resistance at 37 °C)	[78]
Tobacco (22–28 °C)	*R. solanacearum*	Bacterial wilt	C	28, 37 (high humidity >80%)	*ITP5*, *trans*-zeatin	(+)	Cytokinin-mediated resistance (ETI, PTI)	[74]
**Fungi**								
Rice (21–28 °C)	*Magnaporthe oryzae*	Blast	C	38, 35	Several *R* gene loci	(−)	Reduced resistance (QDR)	[79]
	*M. oryzae*	Blast	C	38, 35	*Pik-h*, *Pita2*, *Pii*, *Pi9*, *Piz-t*	(+)	Enhanced tolerance (QDR)	[79]
	*M. oryzae*	Blast	C	38, 35	*Pi54*	(+)	Enhanced tolerance (ETI)	[80]
	*M. oryzae*	Blast	C	14, 22, 28, 33	*MYC2*, *MYC22*, *CEBiP*	(−)	Suppression of JA signaling	[81]
	*M. oryzae*	Blast	F	≥35, 42 (60 h)	*SGS3a/b*, *ARF3a/b*, *ARF3la/lb*	(±)	*SGS3a/b*: (+HS) and (−Pathogen resistance). *ARF3a/b*, *ARF3la/lb*: (−HS) and (−Pathogen resistance).	[72]
	*Rhizoctonia solani*	Sheath blight	T-FACE	Ambient and +2	Malondialdehyde (MDA) levels	(−)	Reduced tolerance	[82]
	*Fusarium fujikuroi*	Bakanae	C	22/18, 26/22, 30/26 (day/night)	*prx98*, *MAPKK*, *GLP 8-7*, *NDR1/HIN1 13*	(−)	Reduced tolerance	[83]
Chickpea (10–22 °C)	*Macrophomina phaseolina*	Dry root rot	C	22/10, 35/25 (day/night)	*-*	(−)	Reduced tolerance	[84]
	*M. phaseolina*	Dry root rot	C	25, 35	*Endochitinase*, *CHI-III*	(−)	Reduced tolerance (PR proteins)	[85]
Coffee (18–21 °C)	*Hemileia vastatrix*	Leaf rust	C	23/18, 27/22 (day/night)	Shade and high nitrogen (N)	(−)	Reduced tolerance under HS; shade and high N enhanced tolerance	[86]
American chestnut (16–27 °C)	*Phytophthora cinnamomi*	Root rot	F	<10, >12 (soil temp.)	Process-based Forest landscape model, biomass	(−)	Reduced tolerance	[87]
Chestnut (16–27 °C)	*P. cinnamomi*	Ink disease	C	30, 35, 45	Secondary metabolite compounds	(−)	Reduced resistance (metabolic aspects)	[88]
Common wheat (15–25 °C)	*Puccinia graminis* f. sp. *tritici*	Stem rust	C	19, 26	*Sr6*	(−)	Reduced resistance (ETI)	[89,90]
	*Blumeria graminis* f. sp. *tritici*	Powdery mildew	C	15, 25	*Pm1*, *Pm8*, *Pm4a*, *Pm4b*	(+)	Stable or enhanced resistance (QDR)	[91]
	*B. graminis* f. sp. *tritici*	Powdery mildew	C	15, 18, 22, 26, 30	*Sr14*, *Sr9b*	(+)	Reduced symptoms (ETI)	[92]
	*Puccinia recondita* (*P. triticina*)	Leaf rust	C	15, 25	Several *R* gene loci	(+)	Stable resistance (QDR)	[93]
	*P. recondita* (*P. triticina*)	Leaf rust	C	18, 22, 25	*LrZH22*	(+)	Enhanced tolerance (QDR)	[94]
	*Puccinia striiformis* f. sp. *tritici*	Stripe rust	C	4–20, 10–30 (diurnal)	Several *R* gene loci	(+)	HTAP tolerance (QDR)	[95]
	*P. striiformis* f. sp. *tritici*	Stripe rust	C	10, 15, 20	*TaXa21*	(+)	HTSP tolerance (PTI, QDR)	[96,97]
	*P. striiformis* f. sp. *tritici*	Stripe rust	C	4–20, 10–30 (diurnal)	*Yr79*	(+)	HTAP tolerance (QDR)	[98]
	*P. striiformis* f. sp. *tritici*	Stripe rust	C	4–20, 10–30 (diurnal)	*Yr62*	(+)	HTAP tolerance (QDR)	[99]
	*P. striiformis* f. sp. *tritici*	Stripe rust	C	4–20, 10–30 (diurnal)	*Yr59*	(+)	HTAP tolerance (QDR)	[100]
	*P. striiformis* f. sp. *tritici*	Stripe rust	C	4–20, 10–30 (diurnal)	*Yr52*	(+)	HTAP tolerance (QDR)	[101]
Durum wheat (15–25 °C)	*P. striiformis* f. sp. *tritici*	Stripe rust	C, F	10–25, 10–35 (diurnal)	*Yr36*	(+)	HTAP tolerance (QDR)	[102,103]
Oat (20–25 °C)	*P. striiformis* f. sp. *tritici*	Stripe rust	C	15, 20, 25, 30	*B*, *E*, *F*, *H*	(−)	Reduced resistance (ETI)	[104]
	*P. striiformis* f. sp. *tritici*	Stripe rust	C	15, 20, 25, 30	*A*, *D*	(+)	Stable resistance (ETI)	[104]
Barley (18–20 °C)	*Bipolaris sorokiniana*	Common root rot, seedling, and head blight, black point, leaf spot	C	20, 49 (water bath for 20 s)	*SOD*, *BI-1*, *DHAR Cyt*, *PR-1b*	(−)	Reduced resistance (ETI)	[105]
	*Blumeria graminis* f. sp. *hordei*	Powdery mildew	C	28, 35 (30 s to 5 days)	*BI-1*, *PR-1b*, *RBOHF2*	(±)	(−) Susceptible lines showed severe symptoms. (+) Resistant line showed stable resistance (ETI)	[106]
	*B. graminis* f. sp. *hordei*	Powdery mildew	C	35, 49	*mlo5*, *Mlg*, *Mla12*	(−)	Reduced resistance (ETI)	[107,108,109]
**Oomycetes**								
*Arabidopsis* (22 °C)	*Peronospora parasitica*	Downy mildew	C	22, 28	*SNC1*	(−)	Compromised resistance (ETI, PTI)	[110]
	*Phytophthora sojae*	Root and stem rot	C	25, 33	Several *R* gene loci	(−)	Reduced resistance (QDR)	[111]
Soybean (19–25 °C)	*P. sojae*	Root and stem rot	C	25, 33	*Rps1-c*, *Rps2*, *Rps5*	(+)	Stable resistance (QDR)	[111]
Pepper	*P. sojae*	Blight and fruit rot	C	25, 37	Multiple *WRKY* and *HSP* genes	(−)	QDR	[112]
Sweet basil (10–25 °C)	*Peronospora belbahrii*	Downy mildew	C	20, >25, 26–31	*-*	(+)	Preheat-treated plants showed stable thermotolerance	[113]
**Viruses**								
Tobacco (22–28 °C)	*Tobacco mosaic virus*	Mosaic	C	28	*Necrosis* (*N*)	(−)	HR suppression (ETI)	[114]
Tobacco (22–28 °C)	*Potato virus X* genes	Mosaic	C	28, 30	*Rx*	(−)	HR suppression (ETI)	[64]
Potato (15–20 °C)	*Potato virus Y*	Mosaic	C	18, 24	*Necrosis y* (*Ny*)	(−)	HR suppression (ETI)	[115]
	*Potato virus Y*	Mosaic	C	22, 28	*PR*, *HSP* genes	(±)	(−) Susceptible lines showed severe symptoms. (+) Resistant line showed stable resistance	[116]
Rice (21–28 °C)	*Rice stripe virus*	Stripe	C	35–37, 38	*Stvb-i*	(+)	Stable resistance	[117]
Tomato (21–27 °C)	*Tomato yellow leaf curl virus*	Leaf curl	C	42–43 (2 h), 40–45/20–25 (diurnal)	*HSFA2*, *HSFB1*, *Hsp17*, *Apx1*, *Apx2*, *Hsp90*	(+)	Stable resistance	[118]
Capsicum (25 °C)	*Capsicum chlorosis virus*	Chlorosis	C	25, 35	RNA silencing-related genes	(+)	Recovery (antiviral RNA silencing activation)	[119]
Common wheat (15–25 °C)	*Wheat streak mosaic virus*	Streak mosaic	C	18, 24	*Wsm1*, *Wsm2*, *Wsm3*	(±)	*Wsm1*/*Wsm2*: (−) Symptoms observed; Only *Wsm3*; (+) Stable resistance	[120]
	*Wheat streak mosaic virus*	Streak mosaic	C	20, 32	Signaling chemical compounds	(−)	Reduced resistance (metabolic aspects)	[121,122]
	*Triticum mosaic virus*	Streak mosaic	C	18, 24	*Wsm3*	(+)	Stable resistance	[120]
Pepper	*Paprika mild mottle virus*	Chlorosis and mottling	C	24, 30	*L ^1a^* locus resistance gene	(−)	HR suppression (ETI)	[123]
	*Tobacco mild green mosaic virus*	Mosaic	C	22, 32	*L ^1^- L ^2^*, *L ^1a^*	(−)	*L ^1^- L ^2^*: HR suppression (ETI) *L ^1a^*: Stable resistance (ETI)	[123]
Common bean	*Bean pod mottle virus*	Bean pod mottle	C	20, 25, 30, 35	*R-BPMV*	(±)	(−) Susceptible lines showed severe symptoms. (+) Resistant line showed stable resistance (ETI)	[124]
**Nematodes**								
Tomato (21–27 °C)	*Meloidogyne incognita*	Roo-knot	C, F	22, 26, 32	*Mi-1*	(−)	Reduced resistance (ETI)	[125,126]
	*M. incognita*	Roo-knot	C, F	22, 26, 32	*Mi-9*	(+)	Stable resistance (ETI)	[125]
	*M. incognita*	Roo-knot	C	25, 32	*Mi-7*, *Mi-8*	(−)	Reduced resistance (ETI)	[127]
	*Meloidogyne javanica*, *M. incognita*, *M. arenaria*	Roo-knot	C	24, 32	*Mi-1*	(−)	HR suppression (ETI)	[128]
Pepper (20–25 °C)	*Meloidogyne* spp.	Roo-knot	C	32, 42	*Me1*, *Me3*	(+)	Stable resistance (ETI)	[129]
**Insects**								
Tomato (14–25 °C)	*Helicoverpa zia* (Corn earworm)	Leaf feeding	C	25/14, 30/18, 35/22	Insect: *GOX* Plant: *TPI*, *PPO*	(−)	Accelerated insect growth, compromised plant growth, and recovery	[130]
Tomato (18–28 °C)	*Manduca sexta* (hornworm)	Leaf feeding	C	28/18, 38/28,	*HSP90*, *COI1*	(−)	Wound-induced JA signaling inhibited growth	[131]
Tomato (19–26 °C)	*Spodoptera littoralis* (noctuid moth)	Leaf feeding	C	20, 25	HS impact on *Trichoderma* sp. and stress responses of tomato by insects	(+)	*T. afroharzianum* T22 enhanced plant resistance against insects fed on detached leaves (no living plant used)	[132]
	*Macrosiphum euphorbiae* (aphid)	Phloem feeding	C	20, 25	HS impact on *Trichoderma* sp. and stress responses of tomato by insects	(+)	*T. afroharzianum* T22 enhanced plant resistance against insects fed on detached leaves (no living plant used)	[132]
Oregano mint (18–24 °C)	*Trialeurodes vaporariorum* (greenhouse whitefly)	Phloem feeding	C	24/18, 45 (5 min)	Volatile compound emissions (lipoxygenase, terpene, benzenoid)	(+)	Heat tolerance, quick recovery from infestation	[133]
Maize (18–22 °C)	*Diabrotica balteata* (banded cucumber beetle)	Root feeding	C	17.8, 20.8 (soil temp.)	Soluble sugars, soluble proteins, benzoxazinoids	(−)	Increased root damage but decreased herbivore survival depending on soil moisture	[134]
Potato (15–25 °C)	*Macrosiphum euphorbiae* (aphid)	Phloem feeding	C	25/15, 30/20	Stomatal conductance	(+)	Reduced survival and fecundity of insects	[135]
Soybean (24–26 °C)	*Aphis glycines* (aphid)	Phloem feeding	C	24–26, 35/20 (diurnal)	Lifespan and fecundity of insects	(+)	Insects fed on detached leaves showed reduced fitness (no living plant used)	[136]

ABA, abscisic acid; ETI, effector-triggered immunity; ISR, induced systemic resistance; HR, hypersensitive response; HTAP, high-temperature adult-plant resistance; HTSP, high-temperature seedling-plant resistance; JA, Jasmonic acid; QDR, quantitative disease resistance; T-FACE system, Temperature by free- air CO_2_ enrichment system. Genetic loci/gene names- *PAD4*, *Phytoalexin Deficient* 4; EDS1, *Enhanced Disease Susceptibility 1*; *RPS2*, *Resistance to P. syringae 2*; *RPS4*, *Resistance to P. syringae* 4; *RPM1*, *Resistance to P. syringae pv Maculicola 1*; *FLS2*, *Flagellin Sensing 2*; *SSL4-SSL5*, *Strictosidine synthase-like 4* and *5*; *MYC2*, *Myelocytomatosis 2* transcription factor gene; *MYC22*, *Myelocytomatosis 22* transcription factor gene; *CEBiP*, *Chitin–elicitor binding protein*; *SGS3a/b*, *Suppressor of gene silencing 3a* and *3b*; *CaIPT5*, *Isopentenyl transferase 5*; *DHAR Cyt*; Cytosolic dehydroascorbate reductase; *SOD*; *Superoxide dismutase*; *BI-1*; *BAX inhibitor-1*; *PR-1b*; *Pathogenesis related-1b*; *RBOHF2*; *Respiratory burst oxidase homologue F2*; *R-BPMV*; *Resistance gene for Bean pod mottle virus*; *mlo5*; *Mildew locus o 5* locus; *Mlg*; *Molecular linkage group* locus; *Mla12*; *barley powdery mildew resistance* locus; *CHI-III*; *PR-3-type chitinase*; *prx98*; *Peroxidase 2 precursor*; *MAPKK*; *MAP kinase kinase*; *GLP 8-7*; *Germin-like 8-7*; *NDR1/HIN1 13*; *NDR1/HIN1-like 13*; *GOX*; *Glucose oxidase*; *TPI*; *Trypsin protease inhibitor*; *PPO*; *Polyphenol oxidase*; *HSP90*; *Heat shock protein 90*; *COI1*; *Coronatine insensitive1*; *SR1/CAMTA3*; *Arabidopsis thaliana signal responsive protein 1/calmodulin-binding transcription activator 3*; *KAN3*; *KANADI (KAN) family protein 3*; *HSF8*; *Heat shock factor 8*; *MLO1*; *Mildew resistance locus O*; *PUB21*; *U-box domain-containing protein 21*.

**Table 2 plants-13-02022-t002:** Plant-beneficial microbes involved in heat stress (HS) tolerance are reported in various crops. AA, amino acids; JA, jasmonic acid; HSP, heat shock protein; HSF, heat shock-responsive transcription factor; SA, salicylic acid; SMs, secondary metabolites; APX, ascorbate peroxidase; SOD, superoxide dismutase; GSH, glutathione; PGP, plant growth promotion; ROS, reactive oxygen species; ABA, abscisic acid, LAX3, like auxin 3; AKT2, potassium channel.

Plant	Plant-Beneficial Microbes	Details	Ref.
Japonica rice	*Paecilomyces formosus*	Reduced endogenous JA and AA, enhanced total protein amounts	[215]
Wheat	*Bacillus amyliliquefaciens*, *Pseudomonas fluorescens*, *Pantoea agglomerans*, *Pantoea agglomerans*	Several common plant protective SMs notably accumulated during HS	[216]
	*Klebsiella* sp.	Protection against salt and HS; reduced ethylene production and regulation of ion transporters	[217]
	*Bacillus cereus*, *Pseudomonas putida*	Increased PGP activity (root, shoot fresh and dry weight, chlorophyll contents) under HS	[218]
	*Bacillus amyloliquefaciens UCMB5113 Abd El-*	Increased plant growth (root, shoot fresh and dry weight, chlorophyll contents) under HS	[216]
	*Pseudomonas brassicacearum*, *Bacillus thuringiensis*, *Bacillus subtilis*	Increase HSP26 protein, GABA, and chlorophyll content, modulated metabolic pathways	[219]
	*Bacillus amyloliquefaciens* UCMB5113, *Azospirillum brasilense*	Augmented antioxidant activities, higher AA and protein content	[220]
	*Bacillus amyloliquefaciens* UCMB5113, *Azospirillum brasilense*	Reduced ROS, higher expression of HSFs and HSPs	[218]
	*Lactobacillus agilis*, *Lactobacillus plantarum*, *Lactobacillus acidophilus*	Increased PGP activity, higher chlorophyll content, enhanced antioxidant levels	[221]
	Thermotolerant bacterial isolates	Increased PGP activity	[222]
Tomato	*Bacillus cereus*	Increased number of flowers and fruits, increased chlorophyll, proline, antioxidants	[223]
	*Paraburkholderia phytofirmans*	Alleviate the harmful effects of HS by PGP	[224]
	*Bacillus safensis*	Significantly improved PGP activity and chlorophyll content, antioxidant enzyme production	[225]
	PGP bacteria synthetic consortium	ACC-deaminase production, ethylene accumulation, PGP	[226]
	*Bacillus cereus*	Reduced ABA, increased SA and antioxidant enzyme activities, and increased APX, SOD, and GSH levels	[227]
Potato	*Pseudomonas* sp. PsJN	Promoted plant growth	[228]
Sorghum	*Azospirillum brasilense* NO40, *Pseudomonas* sp. AKM-P6	Enhanced tolerance in seedling stage to HS	[229,230]
	*Bacillus cereus TCR17*	Reduced ROS stress via upregulation of CAT, APX1, SOD, and HSPs	[231]
	*Pseudomonas* sp. *strain AKM-P6*	Induced stress-related protein production, Preserved membrane integrity, higher levels of sugars, AA, and chlorophyll	[232]
Cabbage	*Bacillus aryabhattai H26-2*, *Bacillus siamensis H30-3*	Higher ABA in leaf, biocontrol activity against soft rot, reduced stomatal opening	[233]
Soybean	*Bradyrhizobium diazoefficiens USDA110*	Survival in starvation	[234]
	*Aeromonas hydrophilla*, *Serratia liquefaciens*, *Serratia proteamaculans*	Increased yields by genistein (antioxidant isoflavone)	[235]
	*Bacillus cereus* SA1	Increased SA, reduced ABA, upregulation of APX, HSP, LAX3, SOD, and AKT2 genes, elevated GSH AA content, increased K gradients	[236]
Legumes	*Rhizobium* sp.	HSP of 63–74 kDa	[237]
Alfalfa	*Shinorizobium meliloti*	Affect symbiosis during HS condition	[238]
Legume	*Rhizobium* sp.	HSP of 63–74 kDa	[215]
Grapevine	PGP bacteria synthetic consortium	Maintained leaf turgidity, lower osmolyte content, enhanced antioxidant mechanisms	[239]
Avocado, tomato	*Pseudomonas*-based consortium	PGP activity	[240]
*Arabidopsis*	*Serendipita indica*	PGP effect	[241]
	*Paraburkholderia phytofirmans* PsJN	PGP and HS amelioration	[242]
Rapeseed, camelina	*Pseudomonas* spp.	PGP activity through carbon reallocation	[243]
Maize	*Bacillus* spp. (AH-08, AH-67, SH-16), *Pseudomonas* spp. SH-29	PGP and HS tolerance during seedling/early vegetative growth	[244]

## Data Availability

Not applicable.

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
