# Peer review of "Heat Stress and Plant–Biotic Interactions: Advances and Perspectives"

_plants, 2024, doi:10.3390/plants13152022_

Round 1

Reviewer 1 Report

Comments and Suggestions for Authors

It is worth noting that the presented review carries the idea of combining the influence of abiotic stress (high temperature) and various biotic factors in the trend of climate change, the consequence of which is global warming. This is the first time in the literature that such a correlation has been attempted for plant systems, especially since, on a global scale, heat stress may be the most significant factor that can fundamentally reshape all existing agricultural models. An undeniable advantage of this review is the detailed presentation of the physiological and biochemical responses of plant systems to each of the biotic stresses in normal and heat shock conditions. It allows for a comparative understanding of the overall response of the plant cell to the effects of such a two-factor stress. Questions and comments that arose during the process of reviewing the manuscript:

1. It is recommended that the “Introduction” section be shortened - it is very detailed in its presentation of commonly known postulates about the negative impacts of global climate change on living organisms, including plants. Please emphasize solely on the heat stress factor in relation to agro ecosystems.

2. Figure 1. - it is recommended to delete figure1A and figure 1C because they carry general information, suitable rather for a graphic abstract. In addition, figure 1B should be summarized in terms of climate zones: the presence of Arabidopsis and the absence of potato is surprising.

3. Figure 2. It is recommended that the authors add a biochemical and molecular biological aspect to the title of the top sub-drawing "Growth, development and physiological challenges".

4. Figure 3. Among the identified stressors, leaf-eating insect pests are absent, only thorn-sucking. Taking into account that they are dominant and cause significant mechanical damage to plants, we would like to see the immune response of the plant at the molecular level.

5. Sections 4 and 5 do not provide enough information on the involvement of PR-proteins and antimicrobial peptides as factors of induced plant immunity. Please expand this point.

6. Figure 4. – a comment similar to point #4.

7. Table 1 does not reflect examples of molecular response of plants under HS and cyst nematodes, only gall nematodes. However, this information is summarized in subsection 5.4 “Plant-Heat Stress-Nematode Interactions”.

8. Figure 5 - the contribution of the endophytic microbiota to improving the immune status of plants should have been shown separately in the figure. The concept of "beneficial microbiota" is very general and in this case aligns more with the definition of "soil microbiota", which is not the same.

Comments on the Quality of English Language

English Language Quality is appropriate.

Author Response

Reviewer #1: It is worth noting that the presented review carries the idea of combining the influence of abiotic stress (high temperature) and various biotic factors in the trend of climate change, the consequence of which is global warming. This is the first time in the literature that such a correlation has been attempted for plant systems, especially since, on a global scale, heat stress may be the most significant factor that can fundamentally reshape all existing agricultural models. An undeniable advantage of this review is the detailed presentation of the physiological and biochemical responses of plant systems to each of the biotic stresses in normal and heat shock conditions. It allows for a comparative understanding of the overall response of the plant cell to the effects of such two-factor stress. Questions and comments that arose during the process of reviewing the manuscript:

Response: Thank you so much for your positive evaluation and valuable comment.

Comment 1. It is recommended that the “Introduction” section be shortened - it is very detailed in its presentation of commonly known postulates about the negative impacts of global climate change on living organisms, including plants. Please emphasize solely on the heat stress factor in relation to agro ecosystems.

Response: Thank you for your critical feedback. While the detailed presentation of the negative impacts of global climate change on agriculture, including plants, is essential for context, we understand the need for brevity and focus. The introduction section emphasizes the connections between climate change, heat stress, biotic, and various environmental factors. By maintaining the detailed Introduction section, we ensure that the manuscript is focused, comprehensive, educational, and scientifically rigorous. We hope Reviewer#1 will agree with this approach, which will enhance the overall quality and impact of the work.

Comment 2. Figure 1. - it is recommended to delete figure1A and figure 1C because they carry general information, suitable rather for a graphic abstract. In addition, figure 1B should be summarized in terms of climate zones: the presence of Arabidopsis and the absence of potato is surprising.

Response: As suggested by Reviewer#1, we have deleted panels 1A and 1C, which are used for a graphic abstract. We have also now included the potato information in the revised Figure 1B. Arabidopsis information is included, considering its significance as a model species in plant biology and its usage in exploring various insights about HS and other stress factors.

Comment 3. Figure 2. It is recommended that the authors add a biochemical and molecular biological aspect to the title of the top sub-drawing "Growth, development and physiological challenges".

Response: Figure 2 is revised as suggested. Thank you.

Comment 4. Figure 3. Among the identified stressors, leaf-eating insect pests are absent, only thorn-sucking. Taking into account that they are dominant and cause significant mechanical damage to plants, we would like to see the immune response of the plant at the molecular level.

Response: As suggested, a representative image of a leaf-eating insect is now included in the revised Figure 3. Also, a discussion about molecular mechanisms underlying plant immune responses to insect pests is added in the main text (Section 4. Plant Responses to Biotic Stressors, Line 278-294). Thank you.

Comment 5. Sections 4 and 5 do not provide enough information on the involvement of PR-proteins and antimicrobial peptides as factors of induced plant immunity. Please expand this point.

Response: Thank you. The revised draft includes details about PR proteins (Line 224-229) and antimicrobial peptides (Line 263-267).

Figure 4. – a comment similar to point #4.

Response: Point included in the Revised Figure 4.

Table 1 does not reflect examples of molecular response of plants under HS and cyst nematodes, only gall nematodes. However, this information is summarized in subsection 5.4 “Plant-Heat Stress-Nematode Interactions”.

Response: We appreciate your careful reading and pointing out this aspect. We agree that Table 1 does not reflect examples of HS and cyst nematode interactions, only gall nematodes. This information is instead summarized in subsection 5.4. Initially, our manuscript included studies on HS and cyst nematodes related to resistant cultivar development or in vitro HS experiments, but these were excluded in later revisions to make the review more precise. We have now revised the text in subsection 5.4 for accuracy and clarity.

Figure 5 - the contribution of the endophytic microbiota to improving the immune status of plants should have been shown separately in the figure. The concept of "beneficial microbiota" is very general and in this case aligns more with the definition of "soil microbiota", which is not the same.

Response: Thank you for raising this point. We intended to depict the generalized concept of beneficial microbiota irrespective of its source. We illustrated it at the bottom part of Figure 5, which seems confusing due to its depiction in soil, and we are sorry for this confusing graphic. Now, we have revised the figure to distinguish it.

Reviewer 2 Report

Comments and Suggestions for Authors

The authors mainly focused on examining recent advances in designing and developing various strategies to address multi-stress scenarios related to HS and biotic factors. The manuscript is written well and informative regarding heat stress and its influence on plant immunity. The study  is suitable for publication after addressing the following points:

1. "On a worldwide scale, the predicted effects of the combined stressors (biotic and abiotic factors) will result in an annual yield loss of more than 50% for major agricultural commodities". It will be better to provide information on the yield losses of agriculture commodities in tabular form.

2. Please remove the old and unnecessary references. Please check it throughout the manuscript.

3.  In line 81 "delves" ?

4. I have found some typing and grammar mistakes, Please check it throughout the manuscript

Author Response

Reviewer #2: The authors mainly focused on examining recent advances in designing and developing various strategies to address multi-stress scenarios related to HS and biotic factors. The manuscript is written well and informative regarding heat stress and its influence on plant immunity. The study is suitable for publication after addressing the following points:

Response: We thank Reviewer#2 for finding our manuscript suitable and valuable to the readers.

Comment 1. - "On a worldwide scale, the predicted effects of the combined stressors (biotic and abiotic factors) will result in an annual yield loss of more than 50% for major agricultural commodities". It will be better to provide information on the yield losses of agriculture commodities in tabular form.

Response: Thank you. We investigated the precise information about yield losses due to biotic and abiotic stressors. The supplementary file provides a summarized Table of suggested facets.

Comment 2. - Please remove the old and unnecessary references. Please check it throughout the manuscript.

Response: Thank you. We have carefully investigated the suggested point. Some old reference works are cited in the review only when the studied aspect is unique and relevant to the topic.

Comment 3. - In line 81 "delves"?

Response: Revised it for clarity. Thank you.

Comment 4. - I have found some typing and grammar mistakes, please check it throughout the manuscript.

Response: The revised version addresses grammatical and typo errors. We have also manually corrected the citation styles. We will attentively address any further points during the proofreading stage. We sincerely appreciate the Reviewer's valuable input and guidance.

Reviewer 3 Report

Comments and Suggestions for Authors

In the review manuscript entitled "Heat Stress and Plant-Biotic Interactions: Advances and Perspectives", Shelake and coworkers summarized the current context of the impact of Heat stress on plant immunity and its effect on plant biotic interaction.

I think this review manuscript is publishable in its current form. On the other hand, I would offer a few comments to improve the manuscript.

I think the authors have cited too many references, could you make a selection of them? 338 references for a review is a lot.

In the chapter Plant Responses to Heat Stress (from line 190), I would suggest the authors to briefly discuss also the role played by HSFA2 in heat stress acclimation responses.

In line 235, please write the entire name for Receptor-like cytoplasmatic kinases before its abbreviation.

I think Figure 4 is not very informative, it's too similar to Figure 3. I would suggest the authors either move it as a supplementary figure or find a different way to represent the impact of HS on PTI and ETI. 

Author Response

Reviewer #3: In the review manuscript entitled "Heat Stress and Plant-Biotic Interactions: Advances and Perspectives", Shelake and coworkers summarized the current context of the impact of Heat stress on plant immunity and its effect on plant biotic interaction.

I think this review manuscript is publishable in its current form. On the other hand, I would offer a few comments to improve the manuscript.

Response: Thank you so much for your positive evaluation and valuable comment.

I think the authors have cited too many references, could you make a selection of them? 338 references for a review is a lot.

Response: Thank you for raising this point. While 338 references may seem extensive, each citation contributes to the depth, credibility, and comprehensive nature of the review. Reducing the number of references could compromise the quality and thoroughness of the work.

In the chapter Plant Responses to Heat Stress (from line 190), I would suggest the authors to briefly discuss also the role played by HSFA2 in heat stress acclimation responses.

Response: Thank you. The role of HSFA2 is discussed in the revised draft.

In line 235, please write the entire name for Receptor-like cytoplasmatic kinases before its abbreviation.

Response: Corrected. Thank you.

I think Figure 4 is not very informative, it's too similar to Figure 3. I would suggest the authors either move it as a supplementary figure or find a different way to represent the impact of HS on PTI and ETI.

Response: Thank you. We agree with Reviewer#3 that Figure 3 and Figure 4 might look similar, but they provide district information. We have now revised Figure 4 with additional details to signify the impact of HS on various aspects of plant immune responses and individual biotic stress agents. Once again, we are thankful for your positive assessment and comments on improving the quality of the manuscript.

Round 2

Reviewer 1 Report

Comments and Suggestions for Authors

Thank to the Authors for the detailed replies and all changes made. I can support most of answers, but I'm not able to agree with the response to the comment #5 ("Although PR proteins are also implicated in abiotic stress responses, their biochemical basis is not entirely known"). PR proteins are known to be involved in plant induced response to biotic stresses mainly, but there are a few papers about molecular aspects of PR proteins. Please expand this subsection with actual references. 

Also Figures 3 and 4 are not modified (the response to the comment #4). Although the authors have added some data regarding leaf-eating insect pests, but they are not displayed on the schemes. It is most important point.

Author Response

Reviewer #1: Thank to the Authors for the detailed replies and all changes made. I can support most of answers, but I'm not able to agree with the response to the comment #5 ("Although PR proteins are also implicated in abiotic stress responses, their biochemical basis is not entirely known"). PR proteins are known to be involved in plant induced response to biotic stresses mainly, but there are a few papers about molecular aspects of PR proteins. Please expand this subsection with actual references.

Response: Thank you for considering and supporting the changes we made in the first round of revision. Regarding PR proteins, we agree with the opinion of Reviewer#1. However, the referred statement is not about the role of PR proteins in biotic stresses (i.e., Although PR proteins are also implicated in abiotic stress responses, their biochemical basis is not entirely known). This statement is about the role of PR proteins in abiotic stresses, and it warrants further studies to establish their biochemical basis.

Additional molecular aspects about the involvement of PR proteins in biotic stresses are further elaborated in the second revision with relevant references. We hope this clarifies and addresses the raised point.

Also Figures 3 and 4 are not modified (the response to the comment #4). Although the authors have added some data regarding leaf-eating insect pests, but they are not displayed on the schemes. It is most important point.

Response: As suggested, Figures 3 and 4 are further revised. Thank you.
